# The nutritional quality of cereals varies geospatially in Ethiopia and Malawi

D. Gashu[1,14], P. C. Nalivata[2,14], T. Amede[3], E. L. Ander[4], E. H. Bailey[5], L. Botoman[2,6], C. Chagumaira[2,5,7,8], S. Gameda[9], S. M. Haefele[8], K. Hailu[1,10], E. J. M. Joy[11], A. A. Kalimbira[2], D. B. Kumssa[5], R. M. Lark[5,7], I. S. Ligowe[2,6], S. P. McGrath[8], A. E. Milne[8], A. W. Mossa[5], M. Munthali[6], E. K. Towett[12], M. G. Walsh[13], L. Wilson[5], S. D. Young[5] & M. R. Broadley[5✉]

Micronutrient deficiencies (MNDs) remain widespread among people in sub-Saharan Africa[1–5], where access to sufficient food from plant and animal sources that is rich in micronutrients (vitamins and minerals) is limited due to socioeconomic and geographical reasons[4–6]. Here we report the micronutrient composition (calcium, iron, selenium and zinc) of staple cereal grains for most of the cereal production areas in Ethiopia and Malawi. We show that there is geospatial variation in the composition of micronutrients that is nutritionally important at subnational scales. Soil and environmental covariates of grain micronutrient concentrations included soil pH, soil organic matter, temperature, rainfall and topography, which were specific to micronutrient and crop type. For rural households consuming locally sourced food— including many smallholder farming communities—the location of residence can be the largest influencing factor in determining the dietary intake of micronutrients from cereals. Positive relationships between the concentration of selenium in grain and biomarkers of selenium dietary status occur in both countries. Surveillance of MNDs on the basis of biomarkers of status and dietary intakes from national- and regional-scale food-composition data[1–7] could be improved using subnational data on the composition of grain micronutrients. Beyond dietary diversification, interventions to alleviate MNDs, such as food fortification[8,9] and biofortification to increase the micronutrient concentrations in crops[10,11], should account for geographical effects that can be larger in magnitude than intervention outcomes.

Globally, more than two billion people are affected by one or more MNDs and the risks of deficiency are greater in sub-Saharan Africa (SSA) than in most other regions[1–3,12]. These MNDs, which are also known as 'hidden hunger', remain a major challenge for achieving the United Nations' Sustainable Development Goal 2 (zero hunger) by 2030[12]. Causes of MNDs include the inadequate intake of micronutrients —in particular, calcium (Ca), iron (Fe), iodine (I), selenium (Se), zinc (Zn) and vitamin A —especially in regions in which diets are dominated by cereals and where access to foods from plant and animal sources that are richer in nutrients is limited[11]. Most cereal grains have inherently small micronutrient concentrations, especially once bran and embryo fractions are removed during milling[10]. Cereal grains also contain large concentrations of anti-nutritional compounds such as phytates (inositol phosphate compounds), which inhibit the absorption of Ca, copper (Cu), Fe, magnesium (Mg) and Zn in the human gut[2,3,5,10].

The prevalence of MNDs must be estimated if policy responses are to be developed in the context of socioeconomic and environmental drivers of food system change[6]. The prevalence of MNDs can be determined from multiple sources of evidence. Biomarkers of status, including micronutrient concentrations or enzyme activities in blood and other tissues, are often used to assess population status[7,12,13]. However, establishing thresholds of sufficiency for biomarkers can be challenging due to variation in the ranges considered to be 'healthy' between demographic groups, physiological buffering and the influence of infection and inflammation, which can have short-term effects on circulating concentrations of micronutrients in the human body[7,14]. Biomarker studies also impose burdens on participants and technical challenges for the collection, storage and analysis of samples, especially in low-income settings. Complementary methods to estimate MND risks include measuring micronutrient intake from composite dietary analyses[15] or, more commonly, from estimates of intake from dietary recall[5], household food consumption and expenditure data[5,16] and food balance sheets[1–3].

Estimates of the prevalence of MNDs from food supply and intake require reliable data on the micronutrient composition of food;

[1]Centre for Food Science and Nutrition, Addis Ababa University, Addis Ababa, Ethiopia. [2]Lilongwe University of Agriculture and Natural Resources (LUANAR), Lilongwe, Malawi. [3]International Crop Research Institute for the Semi-Arid Tropics (ICRISAT), Addis Ababa, Ethiopia. [4]Centre for Environmental Geochemistry, British Geological Survey, Keyworth, UK. [5]School of Biosciences, University of Nottingham, Sutton Bonington, UK. [6]The Department of Agricultural Research Services, Lilongwe, Malawi. [7]Future Food Beacon, University of Nottingham, Sutton Bonington, UK. [8]Sustainable Agriculture Sciences Department, Rothamsted Research, Harpenden, UK. [9]International Maize and Wheat Improvement Center (CIMMYT), Addis Ababa, Ethiopia. [10]Department of Food Science and Applied Nutrition, Addis Ababa Science and Technology University, Addis Ababa, Ethiopia. [11]Faculty of Epidemiology and Population Health, London School of Hygiene & Tropical Medicine, London, UK. [12]World Agroforestry (ICRAF), Nairobi, Kenya. [13]Africa Soil Information Service, Selian Agricultural Research Institute, Arusha, Tanzania. [14]These authors contributed equally: D. Gashu, P. C. Nalivata. ✉e-mail: martin.broadley@nottingham.ac.uk

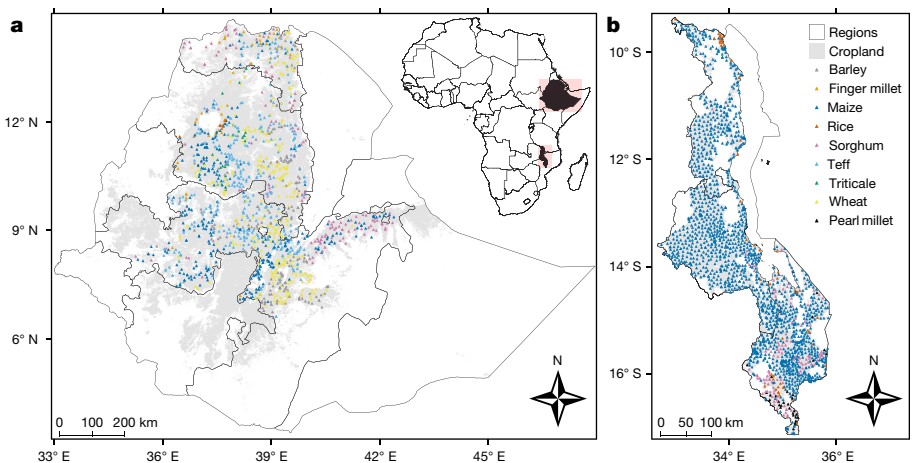

**Fig. 1 | Sampling locations. a**, Ethiopia. **b**, Malawi. Cropland area masks are coloured light grey.

national or international food-composition tables are routinely used in SSA[2,5]. However, data are typically limited to single values for each food–nutrient combination and are therefore of limited value for exploring spatial drivers of micronutrient intake. This is problematic given the widespread consumption of locally produced staple foods such as cereal grains among smallholder farming communities, and given that food micronutrient composition varies within countries[5,17].

Subnational food composition data have not yet been used to inform estimates of the prevalence of MNDs in SSA. This is despite a long history of studies in the fields of medical geology and veterinary sciences, which have linked human and livestock nutritional status to soil and landscape factors. These include studies on the micronutrients I[18], Se[4,18,19], molybdenum (Mo)[20] and cobalt (Co)[21]. Substantial variation in the Ca, Fe, Se and Zn concentrations of staple crops was recently reported from nine districts in Uganda and some of this variation was

associated with soil characteristics[17]. The case for using subnational food composition data to estimate the prevalence of MNDs is strengthened by recent evidence of long-range geospatial variation in grain Se concentration being linked to biomarkers of Se dietary status. In parts of the Amhara region of Ethiopia, nutritionally important variation in the grain Se concentration of teff (*Eragrostis tef* (Zucc.) Trotter) and wheat (*Triticum aestivum* L.) was associated with soil and landscape covariates at distances extending further than 100 km, which is similar to patterns seen among human biomarkers of Se dietary status[22,23]. In Malawi, human Se dietary status was associated with variation in the grain Se concentration in maize (*Zea mays* L.) at localized scales[4,5,15,24–26]. Grain sampling for micronutrient quality has not yet been conducted systematically at wider geographical scales in SSA.

## Grain surveys of cultivated land in Ethiopia and Malawi

Grain micronutrient concentrations in cereal crops are reported here from 1,389 locations in Ethiopia, based on a spatially balanced sample in the Amhara, Oromia and Tigray regions, during the late-2017 and late-2018 harvest seasons. The sampling frame represents a subset of around 354,000 km² of cultivated land–representative of most of the cereal production area in Ethiopia (Fig. 1)–which was constrained to accessible locations that were less than 2.5 km from a known road. At each location, a grain sample and a co-located composite soil sample were taken with the informed consent of the farmer. Grain samples reported from Ethiopia included teff (*n* = 373), wheat (*n* = 328), maize (*n* = 302), sorghum (*Sorghum bicolor* (L.) Moench; *n* = 138), barley (*Hordeum vulgare* L.; *n* = 181) and finger millet (*Eleusine coracana* (L.) Gaertn.; *n* = 39), with a smaller number of triticale (× *Triticosecale* Wittm. ex A. Camus; *n* = 20) and rice (*Oryza sativa* L.; *n* = 8) samples. In Malawi, grain micronutrient concentrations are reported from 1,812 locations, which were sampled during the April–June 2018 harvest season. The Malawi sampling frame represents around 66,000 km² of cultivated land (Fig. 1). Cereal production in Malawi is less diverse than in Ethiopia and grain samples comprised mostly maize (*n* = 1,608) samples, together with sorghum (*n* = 117), rice (*n* = 54), pearl millet (*Pennisetum glaucum* (L.) R. Br.; *n* = 32) samples and a single finger millet sample. Sampling designs, grain analyses and geostatistical methods expanded on those described for part of the Amhara region[22] (Extended Data Figs. 1–9 and Extended Data Tables 1–3).

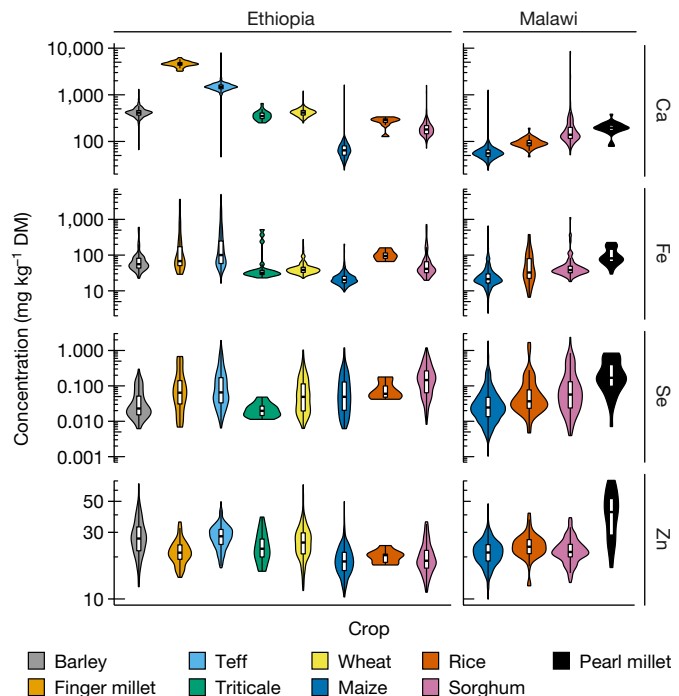

**Fig. 2 | Grain Ca, Fe, Se and Zn concentrations of crop species in Ethiopia and Malawi.** Boxes within the violins show the interquartile range, medians are marked as horizontal lines; whiskers indicate the minimum and maximum values. DM, dry matter.

## Variation between and within crops

Grain Ca, Fe, Se and Zn concentrations varied substantially between and within crop species (Fig. 2 and Extended Data Table 1). Maize typically had

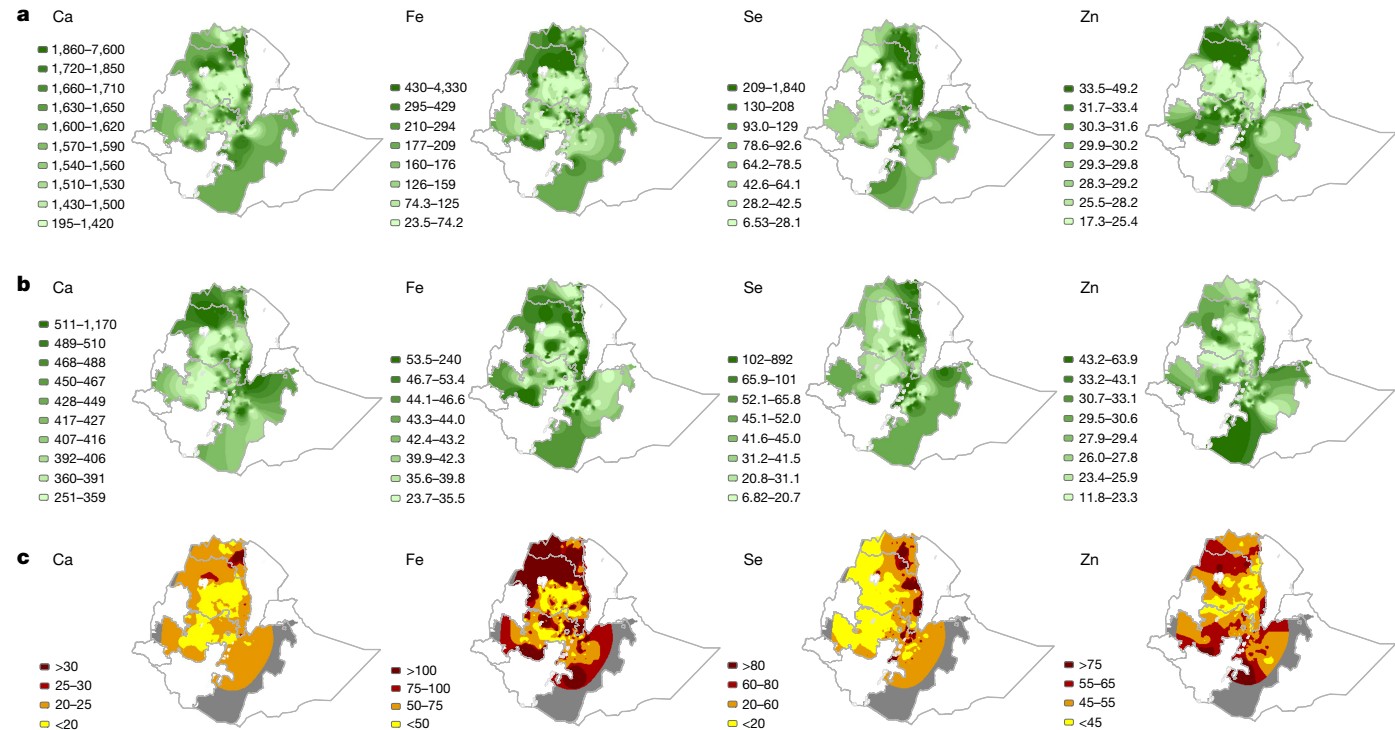

**Fig. 3 | Teff and wheat grain micronutrient concentrations and contribution to dietary supply in Ethiopia. a, b,** Concentrations of micronutrients in teff (**a**) and wheat (**b**) grain. Units are mg kg⁻¹ for Ca, Fe and Zn; μg kg⁻¹ for Se (3 significant figures). **c,** Percentage of micronutrient dietary supply requirement from teff and wheat. The grey shaded area is a mask based on a kriging variance threshold (see Methods). Region boundaries are shown as a grey line.

the lowest concentration of all four micronutrients; people relying on maize-based diets are therefore likely to have the lowest micronutrient intakes. Finger millet is a good potential source of Ca, with a median grain Ca concentration (4,574 mg kg⁻¹; range, 3,203–6,264 mg kg⁻¹) that is almost two orders of magnitude greater than in maize (median, 64.5 mg kg⁻¹) in Ethiopia. The single finger millet sample from Malawi had a grain Ca concentration of 3,564 mg kg⁻¹, which is consistent with the large Ca concentrations that have been reported for finger millet flour from markets in Malawi[26] (range, 2,900–4,700 mg kg⁻¹). A screen of finger millet varieties in India[27] showed a grain Ca concentration range of 1,350–3,120 mg kg⁻¹ (*n* = 49 genotypes), indicating that the trait is conserved across the species. Although the bioavailability of Ca in finger millet will be limited by phytate, as it is for teff, which also had a high grain Ca concentration in Ethiopia (median, 1,473 mg kg⁻¹; range, 46.8–7,925 mg kg⁻¹), fermenting flour to produce injera (a thin flat bread that is commonly consumed in Ethiopia) increases the bioavailability of Ca and other mineral micronutrients by stimulating endogenous phytase enzymes[28].

Within-species variation in grain Ca, Fe, Se and Zn concentrations will arise due to spatial variation in soil and landscape factors, and the effects of extrinsic soil dust. The potential effects of soil dust are pronounced for Fe; soil total Fe concentrations (medians, 92,744 and 28,804 mg kg⁻¹ for Ethiopia and Malawi, respectively) are more than three orders of magnitude larger than median grain Fe concentrations of 20.3 and 21.3 mg kg⁻¹ for maize grain in Ethiopia and Malawi, respectively. Grain concentrations of Se and Zn are less sensitive to soil dust due to the much lower total concentrations of these elements in soils (medians, 0.35 and 0.32 mg kg⁻¹ for Se and 100.1 and 33.7 mg kg⁻¹ for Zn, in Ethiopia and Malawi, respectively).

## Geospatial mapping and dietary contributions

This study focused on mineral concentrations in grain of teff and wheat for Ethiopia and maize for Malawi. These crops were chosen because they comprise a large proportion of the energy intake in national diets and have good spatial coverage in the survey. Grain concentration maps were based on ordinary kriging. Kriging variances—the expected squared error of the predictions—quantify the uncertainties in the maps (Extended Data Figs. 3, 4). The dietary contribution for each crop–nutrient combination was then mapped as a percentage of dietary requirements and visualized on quartile scales from yellow (small) to dark red (large). These calculations used food balance sheets from the Food and Agriculture Organization[29] and assumed estimated average requirement (EAR) thresholds for a representative demographic group—adult women aged 18–24 years eating an unrefined (that is, high phytate) diet[30].

There is spatially dependent variation in grain Ca, Fe, Se and Zn concentrations over large distances in Ethiopia and Malawi (Figs. 3, 4). These observations are likely to be of nutritional importance given that most cereals are grown, milled and consumed locally in Ethiopia[31] and Malawi[5], because they show that the dietary supply and intake of these nutrients varies substantially from one location to another. In Ethiopia, spatial dependencies were seen over distances from 100 to 200 km for the concentration of Ca in teff, for the concentrations of Fe in teff and wheat, and for the concentration of Se in wheat (Extended Data Fig. 2 and Extended Data Table 3). For grain Se and Zn concentrations in teff, and grain Ca and Zn concentrations in wheat, longer-range spatial variation extends beyond 250 km. In Malawi, the spatial dependence of the variation in grain Ca, Fe and Se concentration of maize occurs at distances of 50–80 km. For the concentration of Zn in maize grain in Malawi, more of the variation was attributable to differences over distances that were too short to be resolved by our sampling frame, although the value of the variogram still increased at distances up to 100 km.

In Ethiopia, the Ca concentrations in teff and wheat grain were generally greater in crops sampled from north, northeast and east Tigray region, north and northwest Amhara region and from areas of the Rift

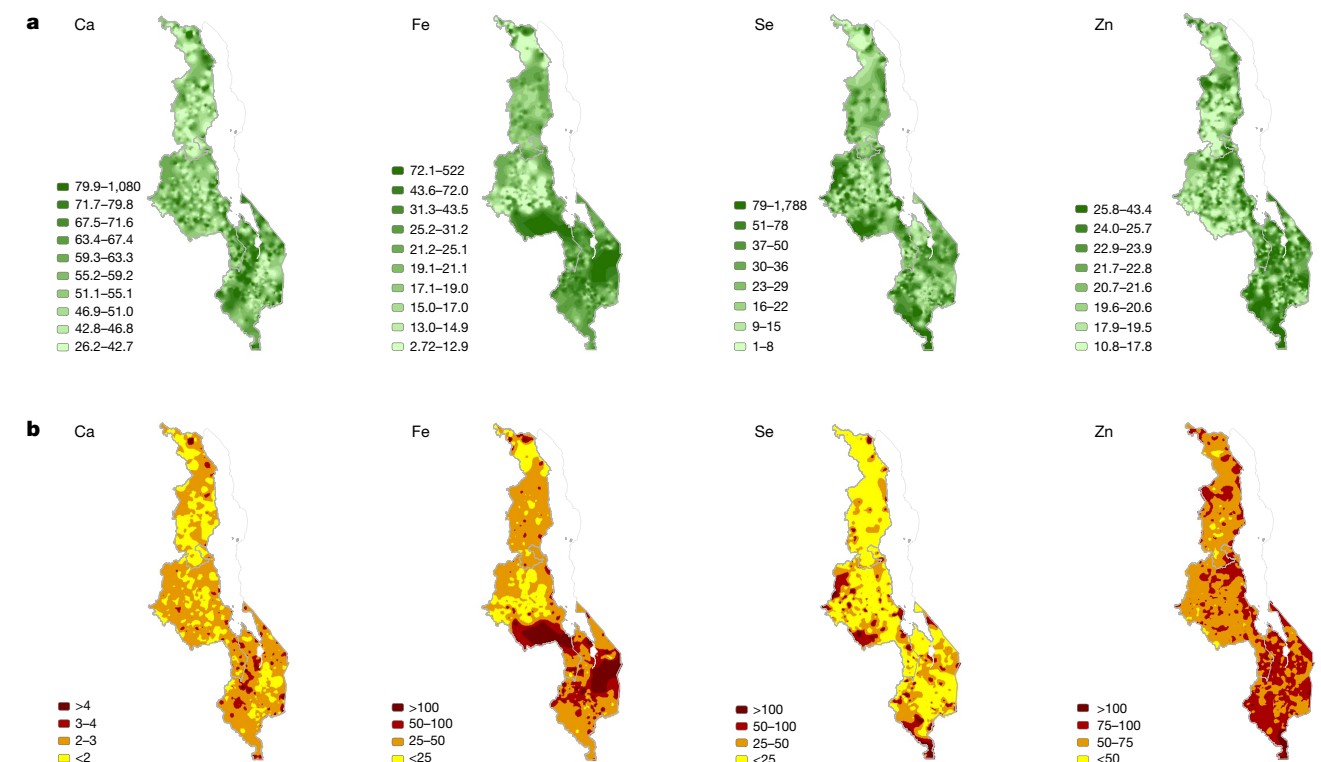

**Fig. 4 | Maize grain micronutrient concentrations and contribution to dietary supply in Malawi. a**, Grain concentration. Units are mg kg$^{-1}$ for Ca, Fe and Zn; µg kg$^{-1}$ for Se (3 significant figures). **b**, Percentage of micronutrient dietary supply requirement from maize.

Valley than crops from most of the Amhara region (Fig. 3). A trend of increasing grain Se concentrations in teff and wheat[22] and increasing Se dietary status of children[31] from the west to the east of the Amhara region has been reported previously. The largest grain Fe and Zn concentrations in teff and wheat were in north Amhara, Tigray and northwest and southwest Oromia regions. In Malawi, maize grain Ca, Se, Zn and—to a lesser extent—Fe concentrations are typically greater in the Shire Valley of the southern region, than in the northern and central regions (Fig. 4). These observations are consistent with observations that crops growing on localized soil types (for example, Vertisols) have greater grain micronutrient concentrations than crops growing on more weathered, acidic soils[4,5,25,26]. As in Ethiopia, geographical areas with the largest maize grain Se concentration in Malawi co-located with areas of Se sufficiency that have been reported in both cross-sectional studies[15] and representative surveillance analyses[24] of human biomarkers of Se dietary status.

Links between grain micronutrient concentration and a dietary outcome were observed for Se, for which there is a reliable biomarker of dietary status[7,13]. Biomarker data for Se for 321 enumeration areas in Ethiopia[23] and 101 enumeration areas in Malawi[24] are available from recent National Micronutrient Surveys. For each enumeration area, the nearest grain sample site for any crop type was identified. In both countries, there is a statistically significant positive relationship between the Se concentration in grain and the Se concentration in serum (Ethiopia) and plasma (Malawi) (Extended Data Fig. 5). Direct evidence of links between the grain concentration of other micronutrients, biomarkers of dietary status and health outcomes remains a major research challenge.

Soil and environmental factors that influence variation in grain micronutrients—for teff, wheat and maize in Ethiopia, and for maize in Malawi—were observed (Extended Data Figs. 6–9 and Extended Data Table 3). For example, soil pH was statistically significantly correlated with grain Se concentration for all crops in both countries (positive relationships). Grain Zn concentration was significantly correlated with soil pH for teff in Ethiopia (negative relationship) and for maize in both countries (positive relationships). Further studies of contrasting responses to soil pH, observed in teff and maize, are needed. However, the generally weak predictive value of soil pH on grain Zn concentration (Extended Data Figs. 7, 9) is consistent with a survey of staple crops in Uganda[17]. Soil organic carbon was significantly correlated with grain Se concentration in wheat in Ethiopia (negative relationship), grain Zn concentration in wheat in Ethiopia (positive relationship) and grain Zn concentration in maize in both countries (positive relationships). Grain micronutrient composition was also significantly correlated with at least one environmental covariate for each micronutrient, crop and country. In Ethiopia, grain Se concentration was negatively correlated with mean annual precipitation for each crop and was positively correlated with mean annual temperature for teff and wheat in Ethiopia and maize in Malawi. Mean annual temperature and topographic index—a measure of soil wetness at a site—were positively correlated with grain Zn concentration for maize in Malawi. Multivariate spatial statistical modelling could be informative about soil and environmental factors that jointly influence the variation of grain micronutrient concentration; however, this requires additional statistical assumptions and is beyond the scope of this study.

In Ethiopia, the proportion of dietary Ca requirements that can be met by the consumption of teff and wheat is much greater than for maize in Malawi, however, it is still likely to be less than 25% of the required amount for most of the population (Fig. 3). In Malawi, maize intake provides less than 3% of dietary Ca requirements for many, despite providing more than 50% of dietary energy requirements (Fig. 4). Cereals provide a greater proportion of the dietary Fe, Se and Zn requirements than Ca in both countries. For example, for some rural households in Malawi, more than 100% of dietary Fe, Se and Zn requirements will be met by eating locally produced maize alone. However, most households will receive less than 25% of Se, less than

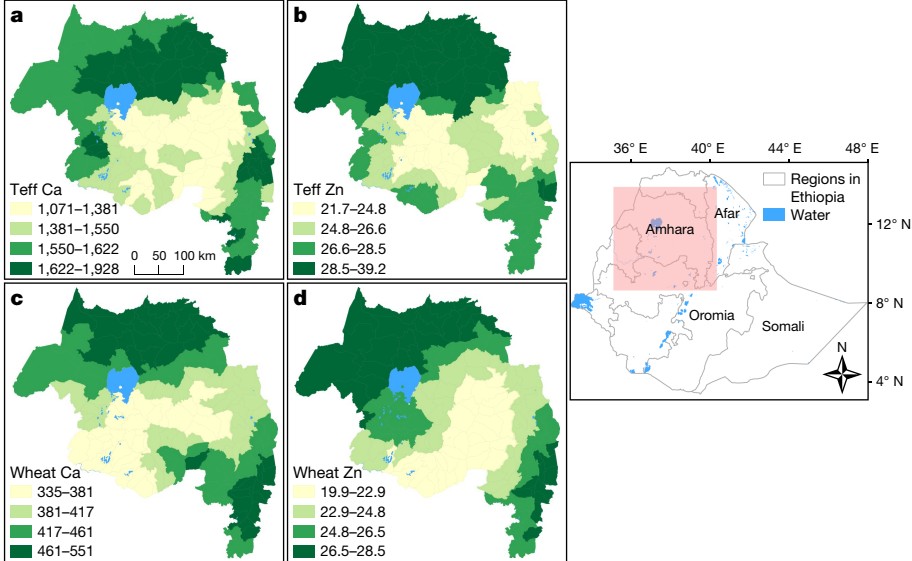

**Fig. 5 | Woreda-level mean grain Ca and Zn concentrations in Amhara region, Ethiopia. a**, Teff Ca. **b**, Teff Zn. **c**, Wheat Ca. **d**, Wheat Zn. Data are in mg kg$^{-1}$. The map on the right shows the analysed area.

50% of Fe and less than 75% of their Zn requirements from a typical consumption pattern for maize.

## Surveillance and interventions

Surveillance of MNDs and food system foresighting activities currently rely on national or international food composition data to estimate micronutrient supply as a proxy for intake[6,7]. Subnational food composition data are not yet used, probably because of the logistical and conceptual challenges in generating data in the required forms. Challenges include sampling crops across large areas during short harvest periods, analysing large numbers of samples and associated data in areas in which access to suitable laboratories and trained personnel is often lacking, and communicating sparse data and associated uncertainties. Spatial dependencies in the grain micronutrient concentrations found in this study can be communicated simply using localized administrative level boundaries. Mean grain Ca and Zn concentrations for teff and wheat show spatial trends when mapped to woreda administrative levels in the Amhara region of Ethiopia using geostatistical models and block kriging (Fig. 5).

Ideally, improved access to diverse diets would alleviate many MNDs[1–6]. However, this is unlikely to be feasible in the shorter term for many people for socioeconomic reasons[6]. A crop micronutrient survey within a country could inform shorter-term interventions to alleviate MNDs. Such interventions include supplementation[32], food fortification[8,9] and biofortification of staple crops through breeding and agronomic approaches[10,11]. For example, Zn-biofortified wheat varieties, which were recently released in India and Pakistan[33,34], have been bred with a target to increase Zn concentrations by 8–12 mg kg$^{-1}$ above a notional baseline grain Zn concentration of 25 mg kg$^{-1}$. Similarly, increases in grain Zn concentration in new Zn-biofortified hybrid maize varieties of 15% in Guatemala (ICTA HB-18 and ICTA B-15) and 36% in Colombia (BIO-MZN01), with target levels of around 30 mg kg$^{-1}$, have been reported[35]. Here, geographical differences in grain Zn concentration, in both wheat (Ethiopia) and maize (Malawi), can exceed these breeding targets. Subnational spatial sources of variation could support priority areas for the release of new crop varieties or micronutrient fertilization strategies and improve impact evaluations of biofortification interventions. The persistence of MNDs is clearly more complex than micronutrient supply in cereal grains[36]. For example, livestock are important components of diverse diets and income;

identifying areas of lower concentrations of micronutrients in forage crops could inform strategies to improve livestock health and production[20,21].

This study did not explore the effects of crop variety (genotype) or farmer management strategies (management), which will contribute substantially to the variation in yield and grain micronutrient concentrations, even over short distances within and between fields of the same farm[37,38]. For example, the preferential use of locally sourced organic materials by smallholder farmers on certain fields can improve the quality and yield of grain micronutrients in maize-based[38] and wheat-based[39] systems in SSA. Temporal variation in environmental (environment) factors, such as the projected decreases in cereal grain micronutrient concentrations due to increased atmospheric $CO_2$—of 6% for Fe and 9% for Zn in wheat by the mid-twenty-first century[40]—and increased leaching of soil Se under higher rainfall[41], should also be considered. However, rising temperatures may compensate for some of these effects in terms of grain micronutrient quality[42]. A better understanding of how the complex interactions between genotype, management and environment drive crop micronutrient quality within diverse, climate-smart farming systems is essential for a more-sustainable global food system.

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

## Methods

### Ethical approval

Grain and soil samplings from farmers' fields were completed in November–December 2017 (for most of Amhara region) and November 2018–January 2019 (Amhara, Oromia and Tigray regions) in Ethiopia, and in April–June 2018 in Malawi. The work involved sampling of grain and soil from farmers' fields and grain stores with the informed consent of the farmers. The work was conducted under ethical approvals from the University of Nottingham, School of Sociology and Social Policy Research Ethics Committee (REC); BIO-1718-0004 and BIO-1819-001 for Ethiopia and Malawi, respectively. These REC approvals were recognized formally by the Directors of Research at Addis Ababa University (Ethiopia) and Lilongwe University of Agriculture and Natural Resources (Malawi), who also reviewed the study protocols.

### Sampling design

The objective of this study was to support the spatial mapping of grain Ca, Fe, Se and Zn concentration of cereal crops. We sought reasonable spatial coverage over the target sample frame and used 'main-site' and 'close-pair' sampling to support the estimation of parameters of the spatial linear mixed model (LMM)[22,43].

In Ethiopia (Amhara, Oromia and Tigray regions), target sample frames were constrained to locations at which the probability of the land being under crop production was ≥0.9 based on predictions produced on a 250-m grid. These predictions were derived from the interpretation of high-resolution satellite imagery by trained observers and by machine learning methods applied to multiple covariates derived from remote sensing data and digital elevation models[44,45]. The sample frame was further constrained to include only those locations from a 250-m grid that fell within 2.5 km of a known road. A map indicating nodes on a 250-m grid (with the same origin as the agricultural land-use grid) that met this requirement was prepared. These constraints may introduce possible biases into predictions made at locations outside the designed sample frame, however, it would not otherwise have been possible to visit all of the sample locations in the time available. Information on the distribution of roads was taken from OpenStreetMap (www.openstreetmap.org). Of a total land area of around 1.1 million km$^2$ in Ethiopia, the total cropland mask represented 354,325 km$^2$, of which 220,467 km$^2$ was within 2.5 km of a known road. In Malawi, the cropland area was determined from the European Space Agency Climate Change Initiative[46]. The agricultural area used was defined as including all raster cells that included the category of 'cropland' in their description. In Malawi, where road access to cropped areas is generally better than in Ethiopia, no constraint to road distance was imposed on sample locations. The mapped cropland areas are shown in Fig. 1.

In Ethiopia, a total of 1,825 primary sample locations were selected a priori, with each 250-m grid node within the sampling frame allocated an equal prior inclusion probability. This was done using the lcube function from the BalancedSampling library for the R platform[47,48]. The lcube function implements the cube method, to enable random sampling according to specified inclusion probabilities while aiming for balance and spread with respect to the specified covariates[49]. Here, inclusion probabilities were uniform across the sample frame and sample locations were selected for spatial balance, which entails that the mean coordinates of sample locations are close to the mean coordinates of all points in the sample frame, and spatial spread, which ensures that the observations are spread out rather than clustered with respect to the spatial coordinates[50]. A subset of 175 of these locations were selected as close-pair sites at which an additional nearby sample was taken to support the estimation of parameters of the spatial LMM[43].

In Malawi, a different sampling method was used to achieve good spatial coverage of a total of 1,710 main-site locations. These included 820 fixed sample points from the 2015/16 Demographic and Health Survey of Malawi[24,51]. The stratify function in the spcosa library for

the R platform[52] divides a sampling domain into Delaunay polygons centred on a set of fixed points and with the remaining polygon centroids selected to partition the domain into approximately equal-area regions. The centroids of the polygons were selected as sample points. An additional 890 sample points were found in addition to the 820 fixed ones, with the stratify function. Once these were obtained, a further 190 locations were selected at random as close-pair sites for an additional nearby sample, as in Ethiopia.

### Field sampling

Sampling was conducted by teams who were trained in standard procedures and risk assessments. Each team planned to visit five main-site locations per day. Main-site locations were loaded onto a computer tablet and printed on paper maps for each team. A team would navigate to the target location, using a handheld GPS device for the last few kilometres. At each sample location, the team would identify the nearest field with a mature cereal crop within a 1-km radius, and sample grain and soil, subject to farmer consent. If a field with a standing mature cereal crop was not apparent, that is, the crop had been harvested, or a non-cereal crop had been grown, the team would ask the farmer to identify a field from which a cereal crop had recently been harvested and stored, and from which a sample could be obtained. If sampling was not possible, then the team would either look beyond a 1-km radius for an alternative location, or abandon the location. At designated close-pair locations, a second field was identified ideally within around 500 m (range, 100–1,000 m) of the main-site location. If a close-pair location could not be found, then a close-pair location would be selected at the next sample location that was not already earmarked for a close-pair sample.

Within a selected field, samples were taken from a 100 m$^2$ (0.01 ha) circular plot. This was centred as close as practical to the middle of the field unless this area was unrepresentative due to disease or crop damage. Five subsample points were located (Extended Data Fig. 1). The first point was at the centre of the plot. Two subsample points were then selected at locations on a line through the plot centre along the crop rows, and two more points on a line orthogonal to the first through the plot centre. Where possible, the central sampling location was fixed between crop rows, and the long axis of the sample array (with sample locations at 5.64 and 4.89 m) was oriented in the direction of crop rows with the short axis perpendicular to the crop rows. A single soil subsample was collected at each of the five subsample points with a Dutch auger with a flight of length of 150 mm and diameter of 50 mm. The auger was inserted vertically to the depth of one flight and the five subsamples were stored in a single bag. Where a mature or ripe crop was still standing in the field, grain samples were taken close to each augering position by a different operator, to minimize further contamination by dust and soil. For maize, a single cob was taken at each of the five points. Maize kernels were stripped from around 50% of each cob lengthways and composited into a single sample envelope for each location. For smaller-grained crops, sufficient stalks were taken so that approximately 20–50% of the sample envelope was filled (dimensions 15 cm × 22 cm), with samples placed grain-first into the sample bag and the stalks were twisted off the grain heads and discarded. If a crop was in field stacks, then a subsample, comprising five cobs for maize, or a representative sample for other crops was taken from each available stack, taking material from inside the stack to minimize contamination by dust and soil (Extended Data Fig. 1). If a crop was in a farmers' store, that is, already averaged from across the field, then a representative sample was taken, while avoiding grain from the store floor if grain was loosely stored and avoiding grain with visible soil or dust contamination.

Photographs at sample locations and of sample bags were recorded for quality assurance along with site GPS locations. In Ethiopia, 1,385 of the 1,389 locations from where grain data are reported had positional uncertainties of ≤8 m as recorded by the GPS. The other four locations

had positional uncertainties of 9–16 m. In Malawi, 1,790 of the 1,812 locations had positional uncertainties of ≤9 m. A further 16 locations had positional uncertainties of 10–17 m, and 6 locations had positional uncertainties of 2,900–5,000 m. We took a decision not to exclude any data based on positional uncertainties for this study. We used robust estimators of the variograms (Extended Data Fig. 2), which are resistant to effects of spatial outliers due to a small number of points being in the wrong position[53] and these models were validated. Any effect of position error on the broad mapped pattern of long-range spatial variation at these scales will therefore be very limited and localized.

## Sample preparation
Whole-grain samples were air-dried in their sample bags. Each sample was then ground in a domestic stainless-steel coffee grinder, which was wiped clean before use and after each sample with a non-abrasive cloth. Whole grains are consumed in many settings, although more-refined fractions—with concomitant losses of more micronutrient-rich bran and endosperm fractions—are often consumed. All preparation was done away from sources of contamination by soil or by dust. A 20-g subsample of the ground material was then shipped to the University of Nottingham. Soil samples were oven-dried at 40 °C for 24–48 h depending on the moisture content of the soil. Preparation took place in a soil laboratory to avoid cross-contamination with grain samples. Plant material was removed from each soil sample, which was then disaggregated and sieved to pass 2 mm. This material was then coned and quartered to produce subsample splits. A 150-g subsample of soil was poured into a self-seal bag, labelled and shipped to the UK for analysis in the laboratories at the University of Nottingham and Rothamsted Research. Soil $pH_{(water)}$ and soil organic carbon (SOC) content were measured using standard methods[22].

## Grain micronutrient analyses
Grain micronutrient analysis methods followed standard methods[54]. Approximately 0.2 g of each ground sample was weighed and digested using a microwave system. For samples collected in the Amhara region in 2017, a Multiwave 3000 48-vessel MF50 rotor (Anton Paar) was used; digestion vessels were perfluoroalkoxy tubes in polyethylethylketone pressure jackets (Anton Paar). Samples were digested in 2 ml 70% trace-analysis-grade $HNO_3$, 1 ml Milli-Q water (18.2 MΩ cm; Fisher Scientific) and 1 ml $H_2O_2$. Settings were: 1,400 W, 140 °C, 2 MPa, for 45 min. For samples collected in 2018–2019, we used a Multiwave Prom 41HVT56 rotor and pressure-activated venting vessels made of modified polytetrafluoroethylene (56-ml 'SMART VENT', Anton Paar). Samples were digested in 6 ml of 70% trace-analysis-grade $HNO_3$. Settings were: 1,500 W, 10 min heating to 140 °C, 20 min holding at 140 °C, and 15 min cooling to 55 °C. Two operational blanks were typically included in each digestion run. Duplicate samples of a certified reference material (Wheat flour SRM 1567b, NIST) were included in approximately every fourth digestion run. Following digestion, each tube was made up to a final volume of 15 ml by adding 11 ml Milli-Q water, then transferred to a 25-ml universal tube (Sarstedt) and stored at room temperature. Samples were further diluted 1:5 with Milli-Q water into 13-ml tubes (Sarstedt) before analysis.

Multi-elemental analysis of grain (Ag, Al, As, B, Ba, Be, Ca, Cd, Cr, Co, Cs, Cu, Fe, K, Li, Mg, Mn, Mo, Na, Ni, P, Pb, Rb, S, Sr, Ti, Tl, U, V and Zn) was undertaken using inductively coupled plasma mass spectrometry (ICP–MS; iCAPQ, Thermo Fisher Scientific). The instrument used a helium collision cell with kinetic energy discrimination to reduce polyatomic interferences. Samples were introduced from an autosampler incorporating an ASXpress rapid uptake module (Cetac ASX-520, Teledyne Technologies) through a perfluoroalkoxy Microflow PFA-ST nebuliser (Thermo Fisher Scientific). Internal standards were introduced to the sample stream on a separate line via the ASXpress unit and included Sc (20 µg l⁻¹), Rh (10 µg l⁻¹), Ge (10 µg l⁻¹) and Ir (5 µg l⁻¹) in 2% $HNO_3$ (Primar Plus grade; Fisher Scientific). An external multi-element calibration standard (Claritas-PPT grade CLMS-2; SPEX Certiprep) was used to calibrate Ag, Al, As, B, Ba, Be, Cd, Ca, Co, Cr, Cs, Cu, Fe, K, Li, Mg, Mn, Mo, Na, Ni, P, Pb, Rb, S, Se, Sr, Tl, U, V and Zn, in the range of 0–100 µg l⁻¹ (0, 20, 40, 100 µg l⁻¹). A bespoke external multi-element calibration solution (PlasmaCAL, SCP Science) was used to create Ca, K, Mg and Na standards in the range of 0–30 mg l⁻¹. B, P and S calibration used in-house standard solutions ($KH_2PO_4$, $K_2SO_4$ and $H_3BO_3$); Ti was determined semiquantitatively. Sample processing was undertaken using Qtegra software (Thermo Fisher Scientific) with external cross-calibration between pulse-counting and analogue detector modes when required. Se was determined separately using a triple quadrupole ICP–MS (iCAP TQ; Thermo Fisher Scientific) using an oxygen cell to mass shift the isotope $^{80}Se$ to $m/z$ 96 ($^{80}Se^{16}O$) to reduce interference from the $^{40}Ar$ dimer. Drift correction was achieved using Rh as an internal standard; calibration used the CLMS-2 multi-element standard (Certiprep).

Analyses were conducted in batches of around 240 samples per run on the ICP–MS instrument (Extended Data Table 1). Individual grain concentration data were corrected for run-specific operational blanks and then converted to concentration on a dry-matter basis. Element-specific limits of detection were reported as 3× the standard deviation of the operational blank concentrations, assuming a notional starting dry weight of 0.2 g of sample material (Extended Data Table 1). For samples for which the grain element concentration was less than the limit of detection, data were removed before the statistical analyses. No adjustment was made for potential contamination of grain samples, for example, with soil dust from the field or store using typical markers (for example, Fe, V). Two sorghum samples, taken from a grain store in Malawi, were excluded from the data analysis based on high concentrations of Ca, Mg and other elements that were considered unlikely to have arisen from soil contamination.

## Statistical analyses
Summary statistics were computed for concentrations of all elements in grain and histograms were examined. It is common for geochemical variables to be log-normally distributed, and the coefficient of skewness of the data was examined using octile skewness as a robust measure of asymmetry of the distribution[55]. The data were analysed on the original scales of measurement (mg kg⁻¹) if the octile skewness was within the interval [−0.2, 0, 2] as described previously[56]. If the octile skewness fell outside this range (with a positive value), and the absolute value of the octile skewness after $log_e$ transformation was smaller than on the original scale, then the data were analysed on the $log_e$ scale (Extended Data Tables 1, 2).

Variograms were estimated for each variable using three estimators (Extended Data Table 2): the standard estimator[57] and two alternatives[58,59]. The two alternative estimators are more robust than the standard estimator to outlying data, be these marginal outliers (apparent on the histogram) or spatial outliers (apparent in local spatial context). The variogram estimates were formed from all pairwise comparisons among observations up to a maximum lag distance (330 km in Ethiopia and 100 km in Malawi), and with lag bins with a width of 10 km. The distance between any two locations (specified by latitude and longitude) was computed as the great circle distance using the distVincentySphere function from the geosphere library of the R platform[47,60]. Exponential variogram functions were then fitted to the estimates by weighted least squares[61]. The exponential variogram was selected because it ensures positive definite covariance matrices for distances on the sphere[62]. Each model was then tested by cross-validation. The selected models are shown in Extended Data Fig. 2.

In cross-validation, each observation was removed and predicted from the remaining observations by ordinary kriging. This was done using each variogram model fitted for each variable to the sets of point estimates computed by the three estimators. The standardized squared prediction error (SSPE) was computed for each observation as the squared difference between the observed and predicted values

divided by the prediction error variance (kriging variance) as computed in the standard ordinary kriging equations[53]. The median value of the SSPE has an expected value of 0.455 in the case of a valid underlying variogram model with normally distributed kriging errors[53]. The standard estimator due to Matheron[57] is more statistically efficient than the robust alternatives, so if the model fitted to these estimates appeared correct from the cross-validation results (the median SSPE is within the 95% confidence interval) then the alternatives were not considered. If the SSPE suggests that the model fitted to Matheron estimates are affected by outliers, then the models fitted to robust estimates were also cross-validated, and one was selected on the cross-validation results[53].

Once a variogram model was selected it was used to compute predictions of the grain concentration (with or without transformation) on nodes of a fine square grid. This was done by ordinary kriging[61]. In ordinary kriging, it is assumed that the mean value of the variable of interest is locally constant, but unknown. An estimate is found that is a linear combination of the observations such that the expected squared error of the prediction (the ordinary kriging variance) is minimized. The kriging variance is also computed as a measure of the uncertainty of the prediction. For a given variogram, the kriging variance reflects the proximity of the location at which a prediction is made to observations of the variable of interest. In the case of variables that had been transformed to logarithms, the prediction at each location was computed by exponentiation of the prediction on the $\log_e$ scale. The resulting back-transformed estimate is a median-unbiased predictor, which is appropriate for variables with a skewed distribution[63]. Note, the kriging variance cannot be meaningfully back-transformed. However, it still indicates how the uncertainty of the predictions varies in space and so it is mapped here on the $\log_e$ scale for transformed variables. The kriging variance maps are shown in Extended Data Figs. 3, 4. In Ethiopia, where the sample sites are irregularly distributed over the mapped area, the kriging variance differs markedly depending on the local sample density.

In Ethiopia, we mapped the percentage of dietary requirement potentially met by the intake of wheat and teff (Fig. 3); similarly, in Malawi for maize (Fig. 4). In Ethiopia, 97.6 g per capita per day of wheat and 89.3 g per capita per day for teff (based on the Food and Agriculture Organization database item, 'other cereals') were used as reference intakes[29]; in Malawi, 342.8 g per capita per day was used as a reference maize intake[29]. Estimated average requirement (EAR) thresholds of 860, 22.4, 45 and 10.2 mg per capita per day for Ca, Fe, Se and Zn, respectively, were chosen as a representative threshold, based on an adult woman aged 18–24 years eating an unrefined (that is, high phytate) diet[30]. These thresholds are similar to other demographic groups. Because a comparable measure of uncertainty to the kriging variance for this derived variable is not available, we defined a mask for Ethiopia, where the sparsity of sampling leads to larger uncertainty compared with Malawi. We considered the Zn concentration in teff, a variable with spatial dependence over long distances, and identified those areas in which the kriging variance for this variable exceeded 75% of the variance of the variable itself because the sampling was sparse. These areas defined the mask used for the maps of the percentage of dietary requirement for all variables and is shown in grey in Fig. 3.

The maps shown in Figs. 3, 4 are made by point kriging—that is, they are predictions of a measurement at an unsampled site. Ordinary kriging can also be used to predict the mean value of a variable across a region or 'block'[61]. To illustrate the communication value of this approach, the mean concentrations of Zn and of Ca in grain (teff and wheat) are shown at the woreda level in the Amhara region (Fig. 5). These were obtained by block kriging of woreda means from the same sample data, and with the same variogram models as used to produce the point kriging predictions.

### Grain–biomarker links

Determining the links between a dietary biomarker of status and grain micronutrient concentration focused here on Se, for which there is a reliable biomarker of dietary status[13]; this is not the case for Zn[7] and the other micronutrients analysed in this study. Data on the concentration of Se in the blood of women of reproductive age were available from micronutrient surveys in Ethiopia (serum[23]) and Malawi (plasma[24]). Mean concentrations were computed for each enumeration area available—321 in Ethiopia and 101 in Malawi—for which the latitude and longitude for each enumeration area centroid were available. For each enumeration area, the nearest grain sample site (regardless of crop) was found. In Ethiopia, the median distance to the nearest grain sample site over all enumeration areas was 17 km. In Malawi, the median distance was much shorter (1.4 km), which is attributable both to the denser crop sampling in Malawi and to the fact that enumeration areas used for the micronutrient survey were targeted for sampling.

The enumeration areas do not comprise a simple independent random sample, so the relation between the serum or plasma Se concentration and the Se concentration in the nearest grain sample could not be quantified by a statistic such as the correlation coefficient. It was therefore studied with an appropriate LMM incorporating a spatially correlated random effect, modelled with a Matérn correlation function[64]. Exploratory analysis indicated that a $\log_e$ transformation of all serum or plasma Se concentrations was necessary for the assumption of normal random effects to be plausible. The observed serum or plasma Se concentration was modelled as a linear function of the concentration in grain at the nearest sample location. Residual maximum likelihood was used to estimate the random effects parameters. The fixed effects were then estimated by weighted least squares[65] along with their standard errors. The evidence against the null hypothesis that the regression coefficient for serum or plasma Se on grain Se concentrations was zero was tested by a log-likelihood ratio test.

Plots of the serum or plasma Se concentration in women of reproductive age, in Ethiopia and Malawi, respectively, show a positive correlation between the variables (Extended Data Fig. 5), which is supported by the statistical models. For Ethiopia, the estimated regression coefficient (log[ng serum Se per ml] and log[mg of grain Se per kg]) was 0.08 with a standard error of 0.02. The null hypothesis that the coefficient was zero is rejected on the grounds of the log-likelihood ratio statistic ($L = 14.48$, $P = 1.4 \times 10^{-4}$). If there was no relationship between the biomarker and the grain Se concentrations, the probability of obtaining an $L$-statistic this large or larger would be very small. Similarly, for Malawi, the estimated regression coefficient was 0.09 with a standard error of 0.03 ($L = 11.56$, $P = 6.7 \times 10^{-4}$).

### Environmental–soil–grain links

Data on the concentration of Se and of Zn in grain (maize in Malawi; teff, wheat and maize in Ethiopia) were extracted along with the corresponding data on soil $pH_{(water)}$ and SOC. Observations for three environmental covariates were also extracted for these same locations: (1) mean annual temperature; (2) mean annual precipitation values from the CHELSA datasets[66,67], which are downscaled to a spatial resolution of 30 s; (3) topographic index values from the 30-s resolution MERIT Digital Elevation Model[68]. The topographic index—which is sometimes called the topographic wetness index—is a measure of the tendency for drainage to accumulate at a site and is widely used as a predictive measure for soil properties. Following exploratory analysis, grain Se concentration data were $\log_e$-transformed to make the assumption of normal random effects plausible; this was not necessary for grain Zn. Measurements of SOC from Malawi showed a marked positive skew and were therefore $\log_e$-transformed.

Data were analysed using a LMM in which a regression-type function of environmental covariates was considered as a fixed effect along with a spatially autocorrelated random effect, as described for the biomarker data. Random-effect parameters were estimated by maximum likelihood, and then the fixed-effect parameters by weighted least squares. The first model was fitted with a constant mean as the only fixed effect. The second model was fitted with mean annual precipitation as a fixed

effect. The evidence for this predictor was evaluated by a log-likelihood ratio test of the second model against the first. The predictor was retained initially if the probability of obtaining a value of $L$ as large or larger than observed under the null hypothesis of no effect of the predictor was <0.05. Additional predictors were then considered in the order of mean annual temperature then topographic index. Finally, the $P$ values were evaluated as the outcome of multiple tests, controlling the false-discovery rate (FDR)[69] at 0.05. The FDR is the expected proportion of incorrectly rejected null hypotheses among all rejected ones. For each micronutrient in each crop and country, the predictors were identified that could be regarded as significant with FDR control at 0.05. The same procedure was followed to produce a comparable model based on the two soil properties, considering first soil pH and then SOC.

Plots of grain Se and Zn concentrations against the environmental covariates and soil properties are shown in Extended Data Figs. 6–9. The evidence for environmental covariates or soil properties as predictors of micronutrient content in the grain is summarized in Extended Data Table 3. Extended Data Table 3 also shows the estimated coefficients, and their standard errors, in separate models for each micronutrient; for maize, teff or wheat in Ethiopia and for maize in Malawi. The predictors in these models are only those that were selected with FDR control.

## Reporting summary

Further information on research design is available in the Nature Research Reporting Summary linked to this paper.

## Data availability

All data are freely available from the corresponding author and available online at https://github.com/rmlark/GeoNutrition.

## Code availability

All code is freely available from the corresponding author and available online at https://github.com/rmlark/GeoNutrition.

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

**Acknowledgements** This work was supported by GeoNutrition projects funded by the Bill & Melinda Gates Foundation (INV-009129) and the UKRI Biotechnology and Biological Sciences Research Council (BBSRC)/Global Challenges Research Fund (GCRF) (BB/P023126/1). The funders were not involved in the study design; the collection, management, analysis, and interpretation of data; the writing of the paper or the decision to submit the paper for publication. The boundaries, denominations and any other information shown on these maps do not imply any judgment about the legal status of any territory or constitute any official endorsement or acceptance of any boundaries on the part of any Government. We acknowledge the contributions made to this research by the participating farmers and field sampling teams. In Ethiopia, field sampling teams were from the Amhara, Oromia and Tigray Regional Bureau of Agriculture. In Malawi, field sampling teams were from the Department of Agricultural Research Services and Lilongwe University of Agriculture and Natural Resources. Training support for field activities in Malawi was facilitated by funding from the Royal Society-UK Foreign, Commonwealth & Development Office (FCDO), under project AQ140000, "Strengthening African capacity in soil geochemistry for agriculture and health". Mineral analytical support was provided by B. Broadley, K. Davis, P. Muleya, S. Vasquez Reina, S. Dunham, J. Carter and J. Hernandez with support from BBSRC Institute Strategic Project Soil to Nutrition (BBS/E/C/000I0310). E.L.A.'s contribution is published with the permission of the Director of the British Geological Survey (UKRI). The CHELSA project is acknowledged for making the downscaled climate data available from https://climatedataguide.ucar.edu/.

**Author contributions** D.G., P.C.N., T.A., E.L.A., S.G., E.J.M.J., A.A.K., R.M.L., S.P.M., E.K.T. and M.R.B. conceptualized the study, and acquired and administered project funding. D.G., P.C.N., E.L.A., E.H.B., L.B., C.C., S.M.H., K.H., E.J.M.J., D.B.K., R.M.L., I.S.L., S.P.M., A.E.M., A.W.M., M.M., M.G.W., L.W., S.D.Y. and M.R.B. contributed to field surveys, laboratory analyses and data analyses, including method development, data interpretation and supervision. R.M.L. conducted geostatistical analyses, including code development. D.B.K. managed the data; C.C., D.B.K., R.M.L. and M.W.G. developed the data visualizations. M.R.B. wrote the primary draft of the paper with editing and reviewing inputs from other authors.

**Competing interests** The authors declare no competing interests.

**Additional information**
**Correspondence and requests for materials** should be addressed to M.R.B.

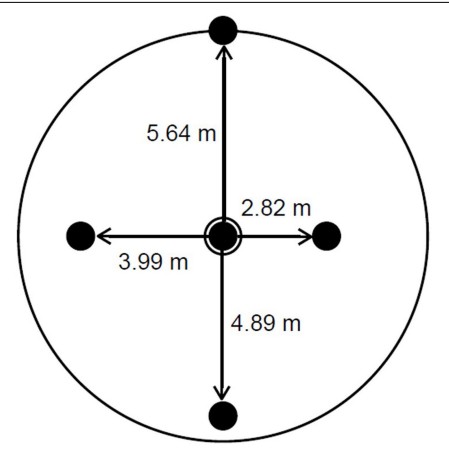

5.64 m

2.82 m

3.99 m

4.89 m

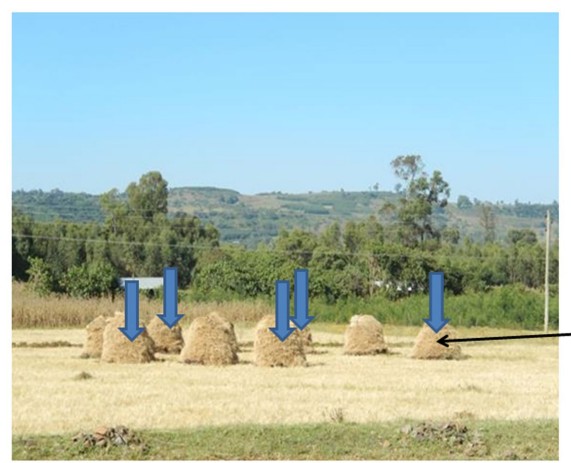

the arrows give
an example of selecting
5 sampling points from
the stacks in the field

collect grain heads
from the centre of
the stack

**Extended Data Fig. 1 | Sampling protocol.** Target layout of the five sample points (black circles) for a standing crop (left) or for a crop harvested in the field (right).

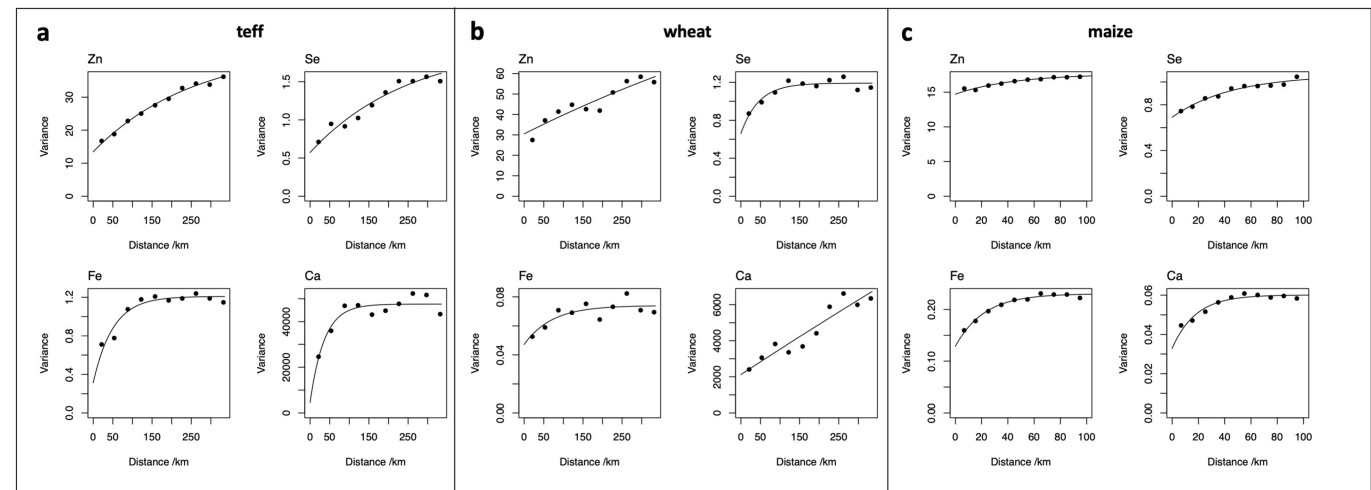

**Extended Data Fig. 2 | Variogram estimates for grain concentrations of Ca, Fe, Se and Zn. a**, Teff in Ethiopia. **b**, Wheat in Ethiopia. **c**, Maize in Malawi.

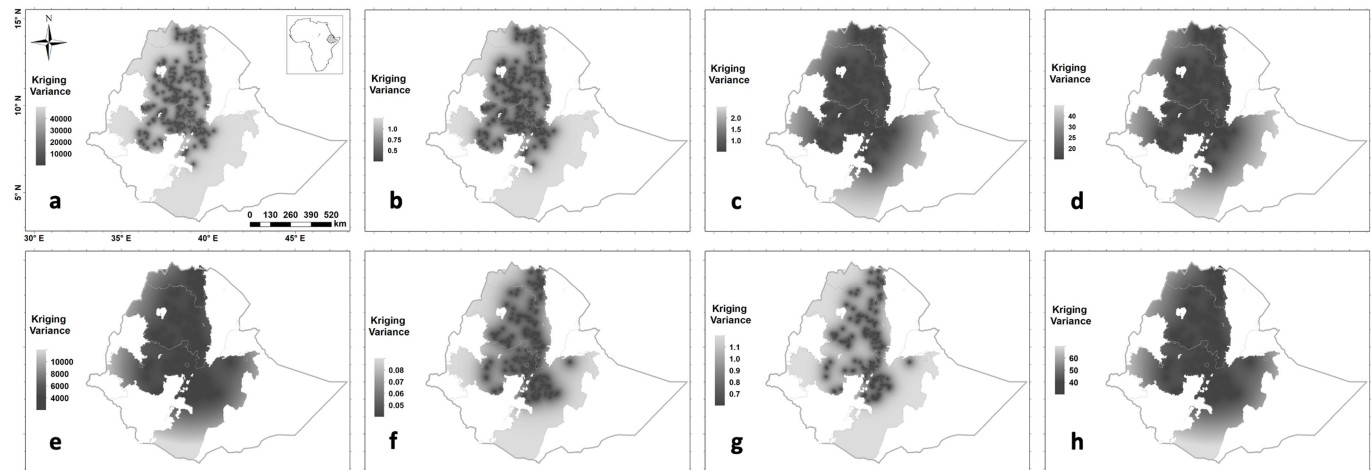

**Extended Data Fig. 3 | Kriging variance maps for grain micronutrient concentrations in Ethiopia. a–h**, Teff (**a**–**d**) and wheat (**e**–**h**) grain concentration of Ca (**a**, **e**), Fe ($\log_e$) (**b**, **f**), Se ($\log_e$) (**c**, **g**) and Zn (**d**, **h**).

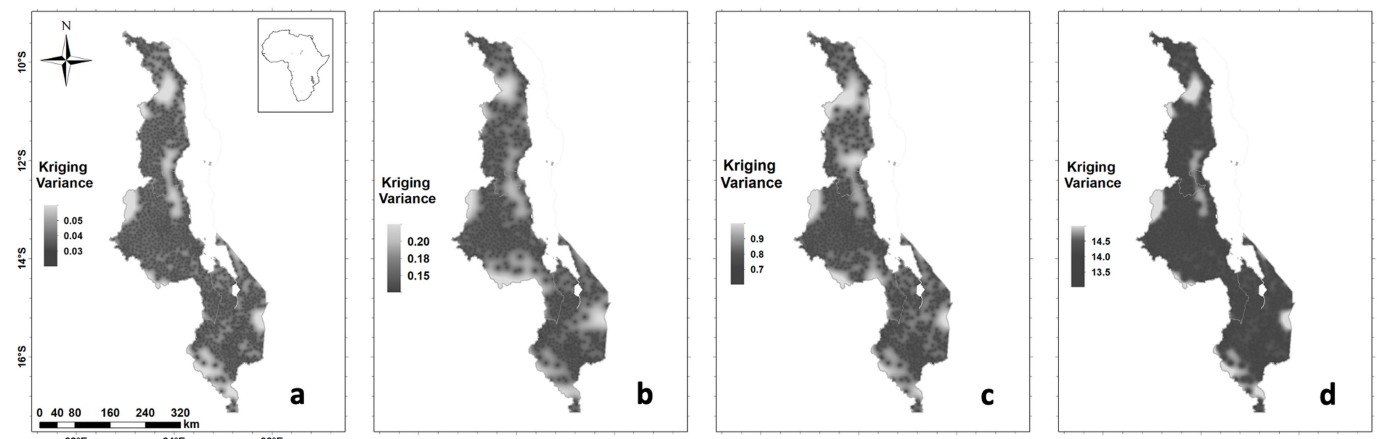

**Extended Data Fig. 4 | Kriging variance maps for maize grain micronutrient concentrations in Malawi. a**, Ca (log$_e$). **b**, Fe (log$_e$). **c**, Se (log$_e$). **d**, Zn.

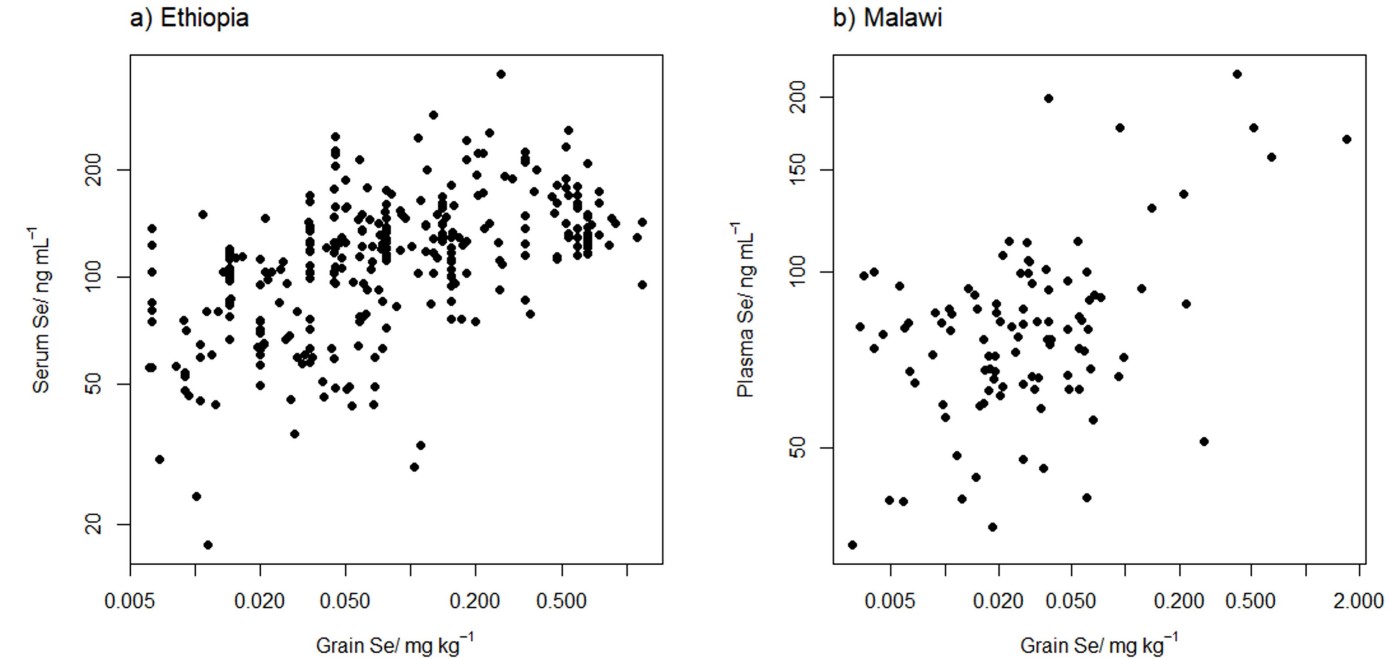

**Extended Data Fig. 5 | Relationships between Se concentration in blood fractions and grain Se concentrations. a**, Ethiopia (serum). **b**, Malawi (plasma).

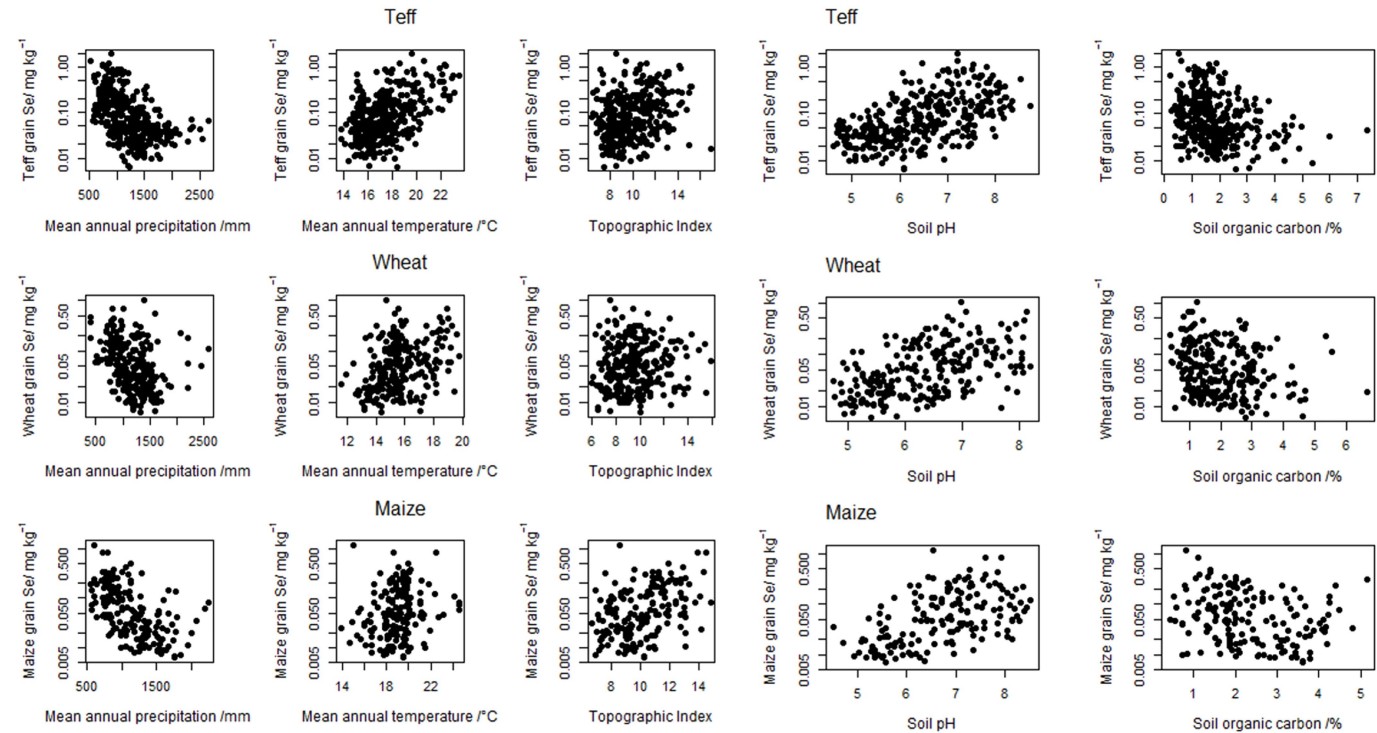

**Extended Data Fig. 6 | Relationships between grain Se concentration and environmental and soil covariates in Ethiopia.** Data for teff (top), wheat (middle) and maize (bottom) are shown.

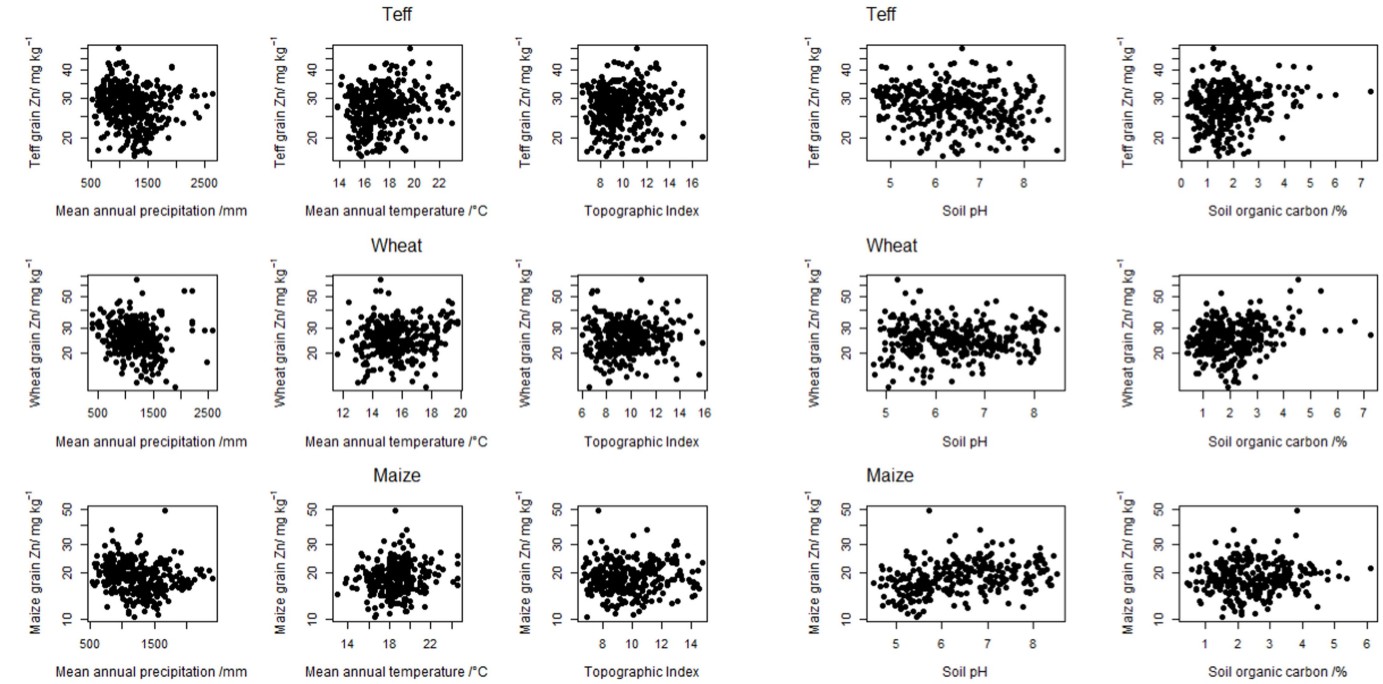

**Extended Data Fig. 7 | Relationships between grain Zn concentration and environmental and soil covariates in Ethiopia.** Data for Teff (top), wheat (middle) and maize (bottom) are shown.

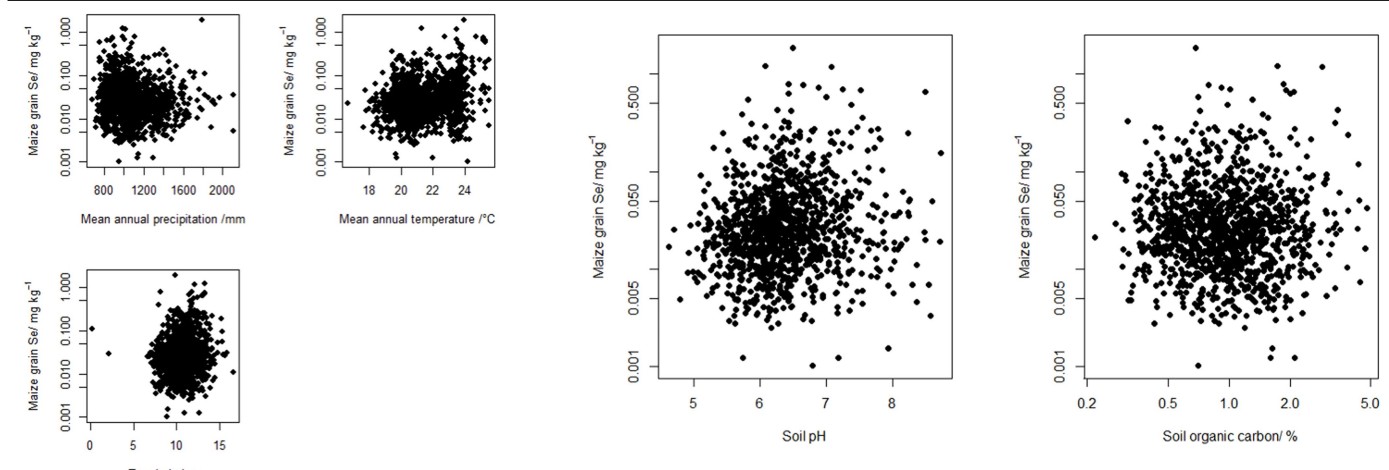

**Extended Data Fig. 8 | Relationships between grain Se concentration and environmental and soil covariates in Malawi.** Data for maize are shown.

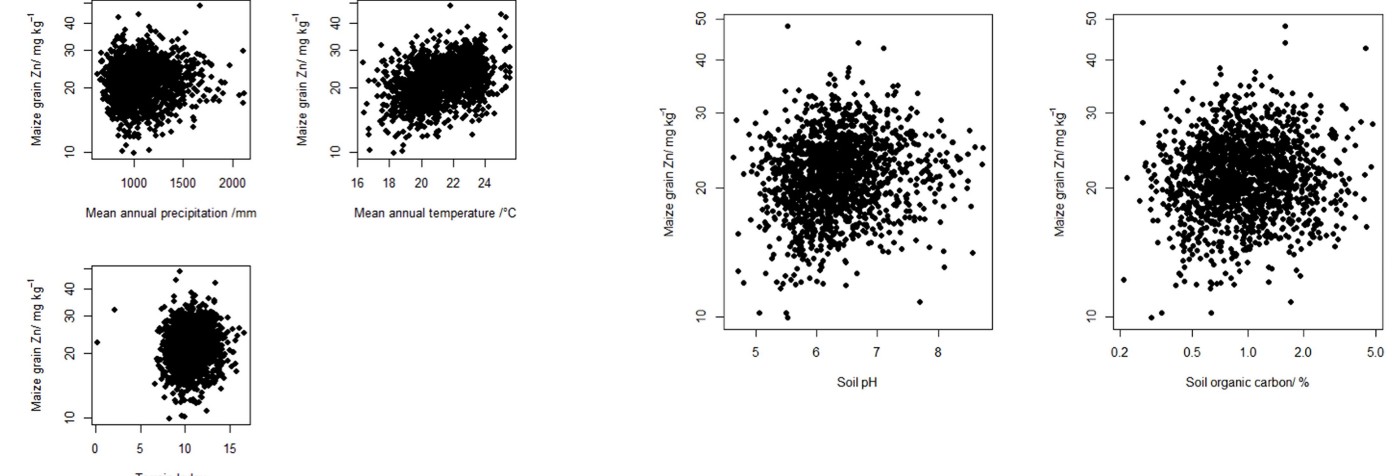

**Extended Data Fig. 9 | Relationships between grain Zn concentration and environmental and soil covariates in Malawi.** Data for maize are shown.

**Extended Data Table 1 | Analytical data for grain micronutrient concentration**

| Survey | Micro-wave runs | ICP-MS runs | Mean LOD or CRM | Ca | Fe | Se | Zn |
|---|---|---|---|---|---|---|---|
| Amhara 2017 | 11 | 2 | LOD (mg kg$^{-1}$) | 9.3 | 5.5 | 0.0094 | 0.9 |
|  |  |  | CRM (%) | 98.3 | 91.3 | 98.4 | 92.6 |
| Ethiopia 2018 | 29 | 5 | LOD (mg kg$^{-1}$) | 12.7 | 8.6 | 0.0195 | 5.0 |
|  |  |  | CRM (%) | 101.6 | 96.5 | 98.6 | 88.8 |
| Malawi 2018 | 51 | 8 | LOD (mg kg$^{-1}$) | 18.3 | 17.9 | 0.0086 | 3.1 |
|  |  |  | CRM (%) | 103.2 | 111.2 | 91.6 | 93.1 |

| Country | Crop | Element | Units | Mean | Median | SD | Skewness | Octile skew |
|---|---|---|---|---|---|---|---|---|
| Ethiopia | Teff | Zn | mg kg$^{-1}$ | 28.23 | 28.1 | 5.44 | 0.35 | −0.05 |
|  | Teff | Zn | log$_e$ mg kg$^{-1}$ | 3.32 | 3.34 | 0.2 | −0.25 | −0.16 |
|  | Teff | Se | mg kg$^{-1}$ | 0.15 | 0.07 | 0.23 | 3.44 | 0.64 |
|  | Teff | Se | log$_e$ mg kg$^{-1}$ | −2.56 | −2.68 | 1.14 | 0.31 | 0.09 |
|  | Teff | Fe | mg kg$^{-1}$ | 297.72 | 101.48 | 543.33 | 4.25 | 0.79 |
|  | Teff | Fe | log$_e$ mg kg$^{-1}$ | 4.93 | 4.62 | 1.1 | 0.94 | 0.39 |
|  | Teff | Ca | mg kg$^{-1}$ | 1507.03 | 1477.36 | 411.33 | 10.16 | 0.09 |
|  | Teff | Ca | log$_e$ mg kg$^{-1}$ | 7.29 | 7.3 | 0.25 | −6.10 | 0.01 |
|  | Wheat | Zn | mg kg$^{-1}$ | 25.97 | 25.37 | 7.15 | 1.22 | 0.1 |
|  | Wheat | Zn | log$_e$ mg kg$^{-1}$ | 3.22 | 3.23 | 0.27 | −0.01 | −0.04 |
|  | Wheat | Se | mg kg$^{-1}$ | 0.09 | 0.05 | 0.12 | 3.51 | 0.61 |
|  | Wheat | Se | log$_e$ mg kg$^{-1}$ | −2.99 | −3.03 | 1.09 | 0.29 | 0.06 |
|  | Wheat | Fe | mg kg$^{-1}$ | 45.15 | 38.59 | 25.94 | 4.41 | 0.42 |
|  | Wheat | Fe | log$_e$ mg kg$^{-1}$ | 3.72 | 3.65 | 0.37 | 1.86 | 0.27 |
|  | Wheat | Ca | mg kg$^{-1}$ | 424.84 | 415.76 | 96.34 | 3.18 | 0.05 |
|  | Wheat | Ca | log$_e$ mg kg$^{-1}$ | 6.03 | 6.03 | 0.19 | 0.95 | −0.05 |
| Malawi | Maize | Zn | mg kg$^{-1}$ | 21.76 | 21.5 | 4.45 | 0.6 | 0.05 |
|  | Maize | Zn | log$_e$ mg kg$^{-1}$ | 3.06 | 3.07 | 0.2 | −0.18 | −0.06 |
|  | Maize | Se | mg kg$^{-1}$ | 0.05 | 0.02 | 0.1 | 8.6 | 0.56 |
|  | Maize | Se | log$_e$ mg kg$^{-1}$ | −3.65 | −3.71 | 1.03 | 0.48 | 0.07 |
|  | Maize | Fe | mg kg$^{-1}$ | 31.34 | 21.29 | 44.95 | 7.91 | 0.46 |
|  | Maize | Fe | log$_e$ mg kg$^{-1}$ | 3.14 | 3.06 | 0.67 | 0.87 | 0.19 |
|  | Maize | Ca | mg kg$^{-1}$ | 59.16 | 55.64 | 35.11 | 25.37 | 0.15 |
|  | Maize | Ca | log$_e$ mg kg$^{-1}$ | 4.03 | 4.02 | 0.28 | 1.28 | 0.01 |

Top, numbers of grain analysis runs for microwave digestion and ICP–MS steps for the grain surveys, including limit of detection (LOD) and certified reference material (CRM) recovery values for Ca, Fe, Se and Zn from wheat flour (SRM 1567b, NIST). Bottom, summary statistics for grain concentrations of Ca, Fe, Se and Zn in teff and wheat in Ethiopia, and maize in Malawi. The selected scale used for analysis is shown in italic.

**Extended Data Table 2 | Cross-validation of variograms**

| Country | Element | Crop | Estimator | $\widetilde{\theta}$ |
|---------|---------|------|-----------|---|
| Ethiopia | Zn | Teff | Matheron | 0.464 |
| | | Wheat | Matheron | 0.436 |
| | Se (log$_e$) | Teff | Matheron | 0.395 |
| | | Wheat | Matheron | 0.509 |
| | Fe (log$_e$) | Teff | Matheron | 0.556 |
| | | Wheat | Dowd | 0.509 |
| | Ca | Teff | Cressie and Hawkins | 0.484 |
| | | Wheat | Dowd | 0.504 |
| Malawi | Zn | Maize | Matheron | 0.413 |
| | Se (log$_e$) | Maize | Cressie and Hawkins | 0.434 |
| | Fe (log$_e$) | Maize | Dowd | 0.512 |
| | Ca (log$_e$) | Maize | Dowd | 0.471 |

Median standardized squared prediction error from cross-validation of the selected variograms for grain concentration of Ca, Fe, Se and Zn in teff and wheat in Ethiopia and maize in Malawi.

**Extended Data Table 3 | Predicting grain micronutrient concentrations based on environmental covariates and soil properties**

| Country | Crop | Nutrient | Predictor | L | p | FDR |
|---|---|---|---|---|---|---|
| Ethiopia | Teff | Se (logₑ) | MAP | 14.03 | $1.8\times10^{-4}$ | ✓ |
| | | | MAT | 24.44 | $7.68\times10^{-7}$ | ✓ |
| | | | TI | 3.96 | 0.047 | ✓ |
| | | | pH | 37.02 | $1.2\times10^{-9}$ | ✓ |
| | | | SOC | 2.63 | 0.105 | |
| Ethiopia | Wheat | Se (logₑ) | MAP | 10.30 | 0.001 | ✓ |
| | | | MAT | 21.43 | $3.66\times10^{-6}$ | ✓ |
| | | | TI | 1.39 | 0.239 | |
| | | | pH | 46.85 | $7.7\times10^{-12}$ | ✓ |
| | | | SOC | 10.38 | 0.001 | ✓ |
| Ethiopia | Maize | Se (logₑ) | MAP | 5.92 | 0.015 | ✓ |
| | | | MAT | 1.14 | 0.285 | |
| | | | TI | 3.05 | 0.081 | |
| | | | pH | 8.68 | 0.003 | ✓ |
| | | | SOC | 0.48 | 0.490 | |
| Ethiopia | Teff | Zn | MAP | 0.80 | 0.371 | |
| | | | MAT | 0.77 | 0.380 | |
| | | | TI | 0.01 | 0.930 | |
| | | | pH | 14.99 | $1.1\times10^{-4}$ | ✓ |
| | | | SOC | 2.42 | 0.120 | |
| Ethiopia | Wheat | Zn | MAP | 0.08 | 0.773 | |
| | | | MAT | 0.36 | 0.547 | |
| | | | TI | 0.01 | 0.935 | |
| | | | pH | 0.54 | 0.463 | |
| | | | SOC | 20.82 | $5.1\times10^{-6}$ | ✓ |
| Ethiopia | Maize | Zn | MAP | 0.49 | 0.485 | |
| | | | MAT | 0.06 | 0.800 | |
| | | | TI | 0.26 | 0.610 | |
| | | | pH | 7.36 | 0.007 | ✓ |
| | | | SOC | 5.68 | 0.017 | ✓ |
| Malawi | Maize | Se (logₑ) | MAP | 0.02 | 0.88 | |
| | | | MAT | 15.29 | $9.2\times10^{-5}$ | ✓ |
| | | | TI | 0.61 | 0.43 | |
| | | | pH | 15.77 | $7.1\times10^{-5}$ | ✓ |
| | | | SOC (logₑ) | 2.24 | 0.13 | |
| Malawi | Maize | Zn | MAP | 1.04 | 0.308 | |
| | | | MAT | 49.20 | $2.3\times10^{-12}$ | ✓ |
| | | | TI | 4.94 | 0.026 | ✓ |
| | | | pH | 14.39 | $1.5\times10^{-4}$ | ✓ |
| | | | SOC (logₑ) | 4.76 | 0.029 | ✓ |

| Country | Crop | Nutrient | Predictor | β | SE | |
|---|---|---|---|---|---|---|
| Ethiopia | Teff | Se (logₑ) | Intercept | −6.14 | | |
| | | | MAP | $-2.7\times10^{-6}$ | $2.5\times10^{-6}$ | |
| | | | MAT | 0.019 | 0.004 | |
| | | | TI | 0.052 | 0.026 | |
| | | | Intercept | −5.40 | | |
| | | | pH | 0.455 | 0.070 | |
| Ethiopia | Wheat | Se (logₑ) | Intercept | −7.09 | | |
| | | | MAP | $-2.5\times10^{-4}$ | $3.3\times10^{-4}$ | |
| | | | MAT | 0.019 | 0.005 | |
| | | | Intercept | −4.97 | | |
| | | | pH | 0.51 | 0.09 | |
| | | | SOC | −0.27 | 0.08 | |
| Ethiopia | Maize | Se (logₑ) | Intercept | −1.54 | | |
| | | | MAP | $-1.0\times10^{-3}$ | $3.6\times10^{-4}$ | |
| | | | Intercept | −5.29 | | |
| | | | pH | 0.38 | 0.12 | |
| Ethiopia | Teff | Zn | Intercept | 36.51 | | |
| | | | pH | −1.40 | 0.35 | |
| Ethiopia | Wheat | Zn | Intercept | 23.48 | | |
| | | | SOC | 1.82 | 0.39 | |
| Ethiopia | Maize | Zn | Intercept | 11.11 | | |
| | | | pH | 1.07 | 0.38 | |
| | | | SOC | 0.68 | 0.28 | |
| Malawi | Maize | Se (logₑ) | Intercept | −6.36 | | |
| | | | MAT | 0.013 | 0.003 | |
| | | | Intercept | −4.72 | | |
| | | | pH | 0.19 | 0.05 | |
| Malawi | Maize | Zn | Intercept | 5.08 | | |
| | | | MAT | 0.09 | 0.01 | |
| | | | TI | −1.56 | 0.07 | |
| | | | Intercept | 18.68 | | |
| | | | pH | 0.59 | 0.17 | |
| | | | SOC (logₑ) | 0.47 | 0.22 | |

Top, significance tests for model parameters. MAP, mean annual precipitation (downscaled); MAT, mean annual temperature (downscaled); pH, soil pH measured in water; SOC, soil organic carbon percentage by mass; TI, topographic index. L is the log-likelihood ratio statistic for the null hypothesis that the named predictor is unrelated to the target variable and p is the corresponding P value. FDR indicates whether the predictor is retained (as indicated by a check mark) with control of the FDR at 0.05. Bottom, estimated coefficients (β) and their standard errors (SE) for selected predictors for grain micronutrient concentrations.

# Reporting Summary

Nature Research wishes to improve the reproducibility of the work that we publish. This form provides structure for consistency and transparency in reporting. For further information on Nature Research policies, see our Editorial Policies and the Editorial Policy Checklist.

## Statistics

For all statistical analyses, confirm that the following items are present in the figure legend, table legend, main text, or Methods section.

| n/a | Confirmed | |
|---|---|---|
| ☐ | ☒ | The exact sample size (*n*) for each experimental group/condition, given as a discrete number and unit of measurement |
| ☐ | ☒ | A statement on whether measurements were taken from distinct samples or whether the same sample was measured repeatedly |
| ☐ | ☒ | The statistical test(s) used AND whether they are one- or two-sided<br>*Only common tests should be described solely by name; describe more complex techniques in the Methods section.* |
| ☒ | ☐ | A description of all covariates tested |
| ☐ | ☒ | A description of any assumptions or corrections, such as tests of normality and adjustment for multiple comparisons |
| ☐ | ☒ | A full description of the statistical parameters including central tendency (e.g. means) or other basic estimates (e.g. regression coefficient) AND variation (e.g. standard deviation) or associated estimates of uncertainty (e.g. confidence intervals) |
| ☒ | ☐ | For null hypothesis testing, the test statistic (e.g. *F*, *t*, *r*) with confidence intervals, effect sizes, degrees of freedom and *P* value noted<br>*Give P values as exact values whenever suitable.* |
| ☒ | ☐ | For Bayesian analysis, information on the choice of priors and Markov chain Monte Carlo settings |
| ☐ | ☒ | For hierarchical and complex designs, identification of the appropriate level for tests and full reporting of outcomes |
| ☒ | ☐ | Estimates of effect sizes (e.g. Cohen's *d*, Pearson's *r*), indicating how they were calculated |

*Our web collection on statistics for biologists contains articles on many of the points above.*

## Software and code

Policy information about availability of computer code

| Data collection | Ref. 44: AfSIS. New cropland and rural settlement maps of Africa. Accessed January 10 2020 from http://africasoils.net/2015/06/07/new-cropland-and-rural-settlement-maps-of-africa/ (2015).<br>Ref 45: Walsh, M. & Wu, W. GeoSurvey data prediction workflows. Open Science Framework (OSF) Repository, https://doi.org/10.17605/OSF.IO/VXC97 (2020).<br>Ref. 46: ESA. Land Cover CCI data [Version 2.0.7, Land Cover data for 2015]. European Space Agency Climate Change Initiative. Accessed November 21 2017 from http://maps.elie.ucl.ac.be/CCI/viewer/download.php (2017).<br>Ref. 47: R Core Team. R: A Language and Environment for Statistical Computing. (R Foundation for Statistical Computing, Vienna, Austria, 2017).<br>Ref 48: Grafström, A. and Lisic, J. BalancedSampling: Balanced and Spatially Balanced Sampling. R Package Version 1.5.2, https://CRAN.R-project.org/package=BalancedSampling (2016). |
|---|---|
| Data analysis | Ref. 47: R Core Team. R: A Language and Environment for Statistical Computing. (R Foundation for Statistical Computing, Vienna, Austria, 2017).<br>Ref. 60: Hijmans, R. J. Geosphere: Spherical Trigonometry. R Package Version 1.5-7, https://CRAN.R-project.org/package=geosphere (2017). |

For manuscripts utilizing custom algorithms or software that are central to the research but not yet described in published literature, software must be made available to editors and reviewers. We strongly encourage code deposition in a community repository (e.g. GitHub). See the Nature Research guidelines for submitting code & software for further information.

## Data

Policy information about availability of data

All manuscripts must include a data availability statement. This statement should provide the following information, where applicable:

- Accession codes, unique identifiers, or web links for publicly available datasets
- A list of figures that have associated raw data
- A description of any restrictions on data availability

All data and code are freely available from the corresponding author (MRB), and available online at [GitHub link to be provided shortly].

# Field-specific reporting

Please select the one below that is the best fit for your research. If you are not sure, read the appropriate sections before making your selection.

☐ Life sciences ☐ Behavioural & social sciences ☒ Ecological, evolutionary & environmental sciences

For a reference copy of the document with all sections, see nature.com/documents/nr-reporting-summary-flat.pdf

# Ecological, evolutionary & environmental sciences study design

All studies must disclose on these points even when the disclosure is negative.

| | |
|---|---|
| Study description | Sampling and analysis of cereal grains from farmers' fields (standing mature crops) and local cereal stores. |
| Research sample | The research sample represents a composited cereal grain sample, together with a co-located soil sample, collected from a standard support frame (see below). Grain samples reported from Ethiopia were from teff (Eragrostis tef (Zucc.) Trotter; n=373), wheat (Triticum aestivum L.; n=328), maize (Zea maize L.; n=302), sorghum (Sorghum bicolor (L.) Moench; n=138), barley (Hordeum vulgare L.; n=181) and finger millet (Eleusine coracana (L.) Gaertn.; n=39), with a smaller number of Triticale (× Triticosecale Wittm. ex A. Camus; n=20) and rice (Oryza sativa L.; n=8) samples. Grain samples reported from Malawi were mostly maize (n=1,608), with sorghum (n=119), rice (n=54), pearl millet (Pennisetum glaucum (L.) R.Br.; n=31), and a single finger millet sample. |
| Sampling strategy | The objective of this study was to support spatial mapping of grain calcium (Ca), iron (Fe), selenium (Se) and zinc (Zn) concentration of cereal crops, across the largest areas of arable cropland in Ethiopia and Malawi that was feasible to sample.<br><br>The sampling strategy used "main-site" and "close-pair" sampling to support estimation of parameters of the spatial Linear Mixed Model (LMM). In Ethiopia (Amhara, Oromia and Tigray Regions), target sample frames were constrained to locations at which the probability of the land being under crop production was ≥0.9 based on predictions produced on a 500-m grid. These predictions were derived from the interpretation of high-resolution satellite imagery by trained observers and by machine learning methods applied to multiple covariates derived from remote sensor data and digital elevation models [Refs. 44,45]. The sample frame was further constrained to include only those locations from a 250-m grid, that fell within 2.5 km of a known road. A map indicating nodes on a 250-m grid (with the same origin as the agricultural land use grid) which met this requirement was prepared. These constraints may introduce possible biases into predictions made at locations outside the designed sample frame, however, it would not otherwise have been possible to visit all of the sample locations across in the time available. Information on the distribution of roads was taken from OpenStreetMap (www.openstreetmap.org). In Malawi, the cropland area was determined from the European Space Agency Climate Change Initiative [Ref. 46]. The agricultural area used was defined as including all raster cells which included the category of 'cropland' in their description. In Malawi, where road access is generally better to cropped areas than in Ethiopia, no constraint to road distance was imposed on sample locations. The mapped cropland areas are shown in Figure 1.<br><br>In Ethiopia, a total of 1,825 primary sample locations were selected a priori, with each 250-m grid node within the sampling frame allocated an equal prior inclusion probability. This was done using the lcube package from the Balanced Sampling library for the R platform [Refs. 47,48]. This implements a cube method, to enable sampling according to specified inclusion probabilities while aiming for balance and spread with respect to specified covariates [Ref. 49]. Here, sample locations were selected for spatial balance, which entails that the mean coordinates of sample locations are close to the mean coordinates of all points in the sample frame, and spatial spread, which ensures that the observations are spread out rather than clustered with respect to spatial coordinates [Ref. 50]. A subset of 175 of these locations were selected as "close-pair" sites where an additional nearby sample would be taken to support estimation of parameters of the spatial LMM [Ref. 43].<br><br>In Malawi, a different sampling method was used to achieve good spatial coverage of a total of 1710 main-site locations. These included 820 fixed sample points from the 2015/16 Demographic and Health Survey of Malawi [Refs. 24,51]. The stratify function in the spcosa package for the R platform [Ref. 52] divides a sampling domain into Delaunay polygons centred on a set of fixed points and with the remaining polygon centroids selected to partition the domain into approximately equal-area regions. The centroids of the polygons were selected as sample points. An additional 890 sample points were found in addition to the 820 fixed ones, with the stratify function. Once these were obtained, a further 190 locations were selected at random as "close pair" sites for an additional nearby sample, as in Ethiopia. |
| Data collection | Sampling was conducted by teams who were trained in standard procedures and risk assessments. Each team planned to visit five main-site locations per day. Main-site locations were loaded onto a computer tablet and printed on paper maps for each team. A team would navigate to the target location, using a GPS for the last few kilometres. At each sample location, the team would identify the nearest field with a mature cereal crop within a 1-km radius, and sample grain and soil, subject to farmer consent. If a field with a standing mature cereal crop was not apparent, then the team would ask the farmer to identify a field from which a cereal crop had |

recently been harvested and stored, and from which a sample could be obtained. If sampling was not possible, then the team would either look beyond a 1-km radius for an alternative location, or abandon the location. At designated close-pair locations, a second field was identified ideally within ~500 m (range 100–1000 m) of the main-site location. If a close-pair location could not be found, then a close-pair location would be selected at the next sample location not already earmarked for a close-pair sample.

Within a selected field, samples were taken from a 100 m2 (0.01 ha) circular plot. This was centred as close as practical to the middle of the field unless this area was unrepresentative due to disease or crop damage. Five sub-sample points were located (Extended Data Figure 1). The first point was at the centre of the plot. Two sub-sample points were then selected at locations on a line through the plot centre along the crop rows, and two more points on a line orthogonal to the first through the plot centre. Where possible, the central sampling location was fixed between crop rows, and the 'long' axis of the sample array (with sample locations at 5.64 and 4.89 m) was oriented in the direction of crop rows with the 'short axis' perpendicular to the crop rows. A single soil subsample was collected at each of the five sub-sample points with a Dutch auger with a flight of length 150 mm and diameter 50 mm. The auger was inserted vertically to the depth of one flight and the five sub-samples stored in a single bag. Where a mature/ripe crop was still standing in the field, grain samples were taken close to each augering position, by a different operator, to minimise further contamination by dust and soil. For maize, a single cob was taken at each of the five points. Maize kernels were stripped from ~50% of each cob lengthways and composited into a single sample envelope for each location. For smaller-grained crops, sufficient stalks were taken so that approximately 20-50% of the sample envelope was filled (dimensions 15 cm × 22cm), with samples placed grain-first into the sample bag and the stalks were twisted off the grain heads and discarded. If a crop was in field stacks, then a sub-sample, comprising five cobs for maize, or a representative sample for other crops was taken from each available stack, taking material from inside the stack to minimise contamination by dust and soil (Figure S2). If a crop was in a farmers' store, i.e. already averaged from across the field, then a representative sample was taken, whilst avoiding grain from the store floor if grain was loosely stored and avoiding grain with visible soil or dust contamination.

Photographs at sample locations and of sample bags were recorded for quality assurance along with site GPS locations. In Ethiopia, 1,385 of the 1,389 locations from where grain data are reported had positional uncertainties of ≤8 m as recorded by the GPS. The other four locations had positional uncertainties of 9–16 m. In Malawi, 1,786 of the 1,807 locations had positional uncertainties of ≤9 m. A further 16 locations had positional uncertainties of 10–17 m, whilst six locations had positional uncertainties of 2,900–5,000 m. We took a decision not to exclude any data based on positional uncertainties for this study. We used robust estimators of the variograms (Extended Data Figure 2) which are resistant to effects of spatial outliers due to a small number of points being in the wrong position [Ref. 53] and these models were validated. Any effect of position error on the broad mapped pattern of long-range spatial variation at national scale will therefore be very limited and localised.

Grain micronutrient analyses followed standard methods [Ref. 54e]; these were conducted in the laboratory by named co-authors of the paper (Mossa, Wilson).

| Timing and spatial scale | Timing: Grain and soil sampling from farmers' fields were completed in November–December 2017 (for most of Amhara Region) and November 2018–January 2019 (Amhara, Oromia, and Tigray Regions) in Ethiopia, and in April–June 2018 in Malawi.<br><br>The sampling frame for Ethiopia represents ~354,000 km2 of cropland, which is most of the cereal production area in Ethiopia. The sampling frame for Malawi represents ~66,000 of km2 area of cropland. The sample locations are provided in Figure 1 of the main paper. |
| --- | --- |
| Data exclusions | No data were excluded based on positional uncertainties. We used robust estimators of the variograms (Extended Data Figure 2) which are resistant to effects of spatial outliers due to a small number of points being in the wrong position [Ref. 53] and these models were validated. Any effect of position error on the broad mapped pattern of long-range spatial variation at national scale will therefore be very limited and localised.<br><br>No adjustment was made for potential contamination of grain samples, e.g., with soil dust from the field or store using typical markers (e.g. Fe, vanadium). Two sorghum samples, taken from a grain store in Malawi, were excluded from the data analysis based on high concentrations of calcium, magnesium and other elements that were considered unlikely to have arisen from soil contamination. These exclusions are reported clearly in the manuscript. |
| Reproducibility | It would be possible to conduct repeated sampling at the same locations given the information reported. |
| Randomization | There were no treatment groups in this study; the sampling design described above embeds randomisation. |
| Blinding | Not applicable to this study. |

Did the study involve field work?    ☒ Yes    ☐ No

# Field work, collection and transport

| Field conditions | The sampling was conducted over a period of several months |
| --- | --- |
| Location | Exact locations for each sample are provided as metadata |
| Access & import/export | The fieldwork was conducted under formal ethical approvals from the University of Nottingham, School of Sociology and Social Policy Research Ethics Committee (REC); BIO-1718-0004 and BIO-1819-001 for Ethiopia and Malawi, respectively. These REC approvals were recognised formally by the Directors of Research at Addis Ababa University (Ethiopia) and Lilongwe University of Agriculture and Natural Resources (Malawi), who also reviewed the study protocols [NB, there are no recognised ethics committee that would cover low-risk sampling of agricultural fields in Ethiopia and Malawi].<br><br>Dried grain samples and dried soil samples was exported from Ethiopia and Malawi to UK, in compliance with all national and international laws, and with sample ownership retained by the lead institution in the country of origin (managed by a collaboration |

agreement). Material was moved into the European Community under Commission Directive 2008/61/EC, and Defra plant health licence number 105352/206483/3 (held by University of Nottingham).

| Disturbance | All field sampling was conducted on arable/cultivated fields, with minimal disturbance. |

# Reporting for specific materials, systems and methods

We require information from authors about some types of materials, experimental systems and methods used in many studies. Here, indicate whether each material, system or method listed is relevant to your study. If you are not sure if a list item applies to your research, read the appropriate section before selecting a response.

## Materials & experimental systems

| n/a | Involved in the study |
|---|---|
| ☒ | ☐ Antibodies |
| ☒ | ☐ Eukaryotic cell lines |
| ☒ | ☐ Palaeontology and archaeology |
| ☒ | ☐ Animals and other organisms |
| ☒ | ☐ Human research participants |
| ☒ | ☐ Clinical data |
| ☒ | ☐ Dual use research of concern |

## Methods

| n/a | Involved in the study |
|---|---|
| ☒ | ☐ ChIP-seq |
| ☒ | ☐ Flow cytometry |
| ☒ | ☐ MRI-based neuroimaging |

