## [Peer Review File · Nature]

Manuscript Title: Cereal micronutrient quality varies geospatially in Ethiopia and Malawi

Editorial Notes: *none*

Reviewer Comments & Author Rebuttals

Reviewer Reports on the Initial Version:

Referee #1 (Remarks to the Author):

Comments

Micronutrient deficiency is an important and widespread problem in Sub-Saharan Africa, especially in areas where a cereal-based diet is dominant. It is also well-known that there is huge geospatial variation in soil quality and therefore in the availability of micronutrients for uptake by the crop and accumulation in the grains. Moreover, there is a wide diversity in traditions of processing the grains in food items and therefore the bioavailability of micronutrients might differ from region to region as well. Finally diets may vary enormously and therefore other sources of micronutrients such as fruits and leafy vegetables vary as well.

A detailed analysis of the micronutrient contents of cereals in Sub-Sahara Africa is therefore most welcome. This paper certainly holds promise as it is based on a very large set of samples in two countries collected according to rigorous sampling techniques and followed by accurate processing and analysis if the samples. I complement the authors with this huge data set and with the thorough geospatial analysis. I have, however, some comments, which by the way, are biased by my personal agronomic and crop-physiological background.

1. As an agronomist, I would not call Ca a micronutrient. In my view it is a macronutrient, at least for a plant.
2. Variation in soil acidification might play a major role in this data set. At least for Ethiopia I have seen soil maps with huge variation in pH levels on short distances which will obviously have a huge impact on uptake of certain micronutrients by the crop. Is it possible to overlay the variation in quality with the soil pH maps?
3. Similarly, there is huge variation in altitude in Ethiopia, also over relatively short distances. It is likely that variation in elevation will have strong effects on micronutrient content because of variation in temperature during grain filling. Temperature will strongly increase grain filling rate and shorten grain filling duration, with the ultimate effect that single-grain weight will become lower when temperature during grain filling will be higher. It would be useful to assess whether the variation in micronutrient content is associated with variation in altitude.
4. Related with point 3 but also related to general agronomic conditions: Thousand grain weight could be an important explaining co-variable both among species and within species. Has that trait been recorded? From a crop physiological and agronomic point of view it would be essential to interpret variation in micronutrient content based on that trait, so the background of the geospatial variation would be much easier interpreted when that trait is known and reported.
5. How does the processing of the grain samples in this research compare with the way the grains are processed for food? Could that be discussed in more detail?
6. Are there enough data collected for a high level of accuracy in Figures 3 and 4? Contrasts in these figures are often very large over very short distances and I wondered whether that had to do with small numbers of observations for specific crops in specific parts of the country.
7. Diet diversification is probably a faster route to solving hidden hunger than breeding for

increased micronutrient levels. This could be discussed in more detail.

8. The statement in lines 263 – 265 on the decline in micronutrient concentration caused by the impact of increased CO₂ in the future is questionable. There have been some alarming papers on that potential effect but in many cases the other factor in climate change (increased temperature) was not taken into account. The increase in temperature compensates for the increase in CO₂ and the outcome of those opposite effects is uncertain and very much depends on the crop and on the micronutrient. We have collected detailed data sets from T-FACE experiments and we could not see a consistent effect on micronutrient contents in the grain of wheat and rice nor in their bioavailability (paper submitted, not published yet). Other research have also reported on micronutrient contents in wheat, rice and soybean grain obtained from T-FACE experiments and concluded that elevated temperature could compensate for elevated CO₂.

See e.g.

Köhler, I. H., Huber, S. C., Bernacchi, C. J., & Baxter, I. R. (2019). Increased temperatures may safeguard the nutritional quality of crops under future elevated CO₂ concentrations. *The Plant Journal*, 97(5), 872-886.

Wang, J., Li, L., Lam, S. K., Liu, X., & Pan, G. (2020). Responses of wheat and rice grain mineral quality to elevated carbon dioxide and canopy warming. *Field Crops Research*, 249, 107753.

9. It is highly likely that better grain yields are associated with lower micronutrient concentrations, but nevertheless, higher yields would allow more food intake and therefore more micronutrient intake despite lower concentrations. Can the authors discuss this as well and can they speculate to what extent the geospatial data on micronutrients are in line with or in contrast to geospatial variation in yield?

Minor edits:

Typo line 404: famers should be farmers

Lines 804-805: sentence incomplete.

Referee #2 (Remarks to the Author):

The authors present the micro-nutrient concentrations of cereal crops in 2 densely populated countries of sub-Saharan Africa and relate this information to limitations in the intake of such essential elements. The presented work is original, the methodology used appropriate, the data presentation of high quality, and the statistics used state-of-the-art. The text is well-written and the content of the various sections in line with the journal requirements.

While the sampling frame of the study is well laid out and the data presentation impressive, there are a number of limitations that would require extra efforts to have an unequivocal understanding of how micro-nutrient concentrations in crops affect people's health status:

1. Explanatory value: The study presents data in a descriptive way and identified spatial dependencies acting over a maximal distance but does not provide more insights in what drives micro-nutrient concentrations. While the authors tool paired grain-soil samples, there is no further mention of the soil information in the paper. We also know that in both countries, fertilizer use is happening with some of these products having Ca and/or Zn included. The impact of this management intervention is not presented. In short, my feeling is that the presented work is a really excellent and necessary first step but more work is needed to add value to our understanding of what drives micro-nutrient concentrations.

2. Varietal differentiation: We know that different varieties of the targeted cereals very likely have varying micro-nutrient uptake potentials and this is not explored by the authors. It would be key to zoom in on this issue, which could explain some of the unexplained observed variation.

3. Variation over time: It would be good to understand how seasonal variations or variations in

crop and soil management practices over time affect micro-nutrient concentrations. It is likely that concentrations are not constant over time.

4. Linkages between measured micro-nutrient concentrations and dietary supply (Figs 3 and 4): Estimation of the dietary supply is done in a 'quick and dirty' way and does not include information on total availability of grain, intake of other food groups, nor the bio-availability of the micro-nutrients for the different crops. Therefore, in their current form, the information presented is of limited value and certainly not sufficient for policy formulation.

4.

Based on above observations, I have

Referee #3 (Remarks to the Author):

This manuscript reports an enormous body of work which is quite staggering in its coverage – the coordination and logistical effort that has been taken to collect and process such a large number of samples from such a wide geographic area in both Ethiopia and Malawi is truly impressive. The sampling frame and statistical approach used to locate the samples and the methods used for sampling and analysis look to be well thought through and robust. The data analysis and presentation is clear, although the figures and maps are too small to read the detail. I suggest that large versions of all maps are published online to allow detailed scrutiny of the maps by readers.

My concern is whether the authors have truly established that, as stated in lines 46-47, the “geospatial variation is nutritionally important”. This point is discussed in the paragraph from lines 197-209. We are told that the areas where larger concentrations of selenium are found in the grain co-locate with areas where better Se status of children has been reported. So essentially the paper relies on previously published information to state that the geographical patterns of micronutrient distribution are meaningful for human nutrition and health. As a reader there appears to be no way to check this – the paper certainly has demonstrated large differences in micronutrient content of cereal grains across geographic areas but I wonder whether the authors have any means to demonstrate more clearly the links with human nutrition.

It is not clear what the authors are recommending in terms of “surveillance” – would it be sufficient to conduct a one time national survey such as this to establish which regions are of concern in terms of restricted concentrations of micronutrients in grain? Or would more regular sampling be needed? Given the huge amount of work involved to conduct such a spatial analysis it seems unlikely this would be feasible on any regular basis. Further all samples were shipped to the UK for analysis – is such analysis feasible within each country or would substantial investments be needed in laboratory infrastructure and training?

Little is made of the spatial dependence of concentrations of different nutrients in grain. If these are related to soil type – why is there different spatial dependence for different nutrients? Surely this is one of the more interesting outcomes of the study? Are the differences among soil types masked by differences in management (use of organic and mineral fertilisers?). It is well established that soil fertility varies enormously over very short distances (50-100 m) within farms in sub-Saharan Africa and in particular in Ethiopia there are strong catenary effects on soil type/soil fertility over distances of a few km. These are very different scales compared with those mapped at subnational scale and discussed in the paper (see L 183-195).

I find the final paragraph of the paper is rather weak and needs rethinking to emphasise the importance of this study.

I see value in the research and am keen to see the paper published. If the authors can address the

points raised above this will strengthen the paper enormously.

Specific comments

Please make it clear from the start that the article is about human nutrition and not plant nutrition

L 47 – representative – of what? Spatially representative? – this is defined later in L 117 as

“systematic spatial sampling” but this needs to be defined earlier.

L 55 – in what way are food fortification and biofortification “food system interventions”?

“geographical effects which can be larger than planned intervention outcomes”

L 61 – greater than what?

L 80 – in the human body?

L 88 delete “having”

L 91 – single values from tables – these are not usually single data points?

L 91 – confusing punctuation

L 96 do not use “a.k.a.” – delete or define “geonutrition” – this is not a commonly used term and unnecessary jargon?

L 96/97 – is this true? The paper cites several studies that show such links – the statement needs clarification

L 109 – nationally representative? See comment above.

L 120 – composite not composited

L 205 – vertisols are NOT alluvial soils – this needs correcting!! Vertisols are common in Ethiopia but much less common in Malawi

L 264 – there is considerable debate as to whether elevated CO₂ will reduce cereal grain micronutrient concentrations due to the compensating effects of increased temperatures. This statement needs more nuance.

L 423 – remote sensing?

L 434 – turn around “is generally better to cropped areas” to read “to cropped areas is generally better”

L 530 – in this paper? Rather than study.

Figs 3 and 4 – the keys are too small to read. I do not see a grey mask in Fig. 4?

Referee #4 (Remarks to the Author):

This paper marshals an abundance of new field data to tackle an important and under-researched problem: intra-national spatial variation in micronutrient (MN) density of staple foods in low-income rural areas where micronutrient deficiencies are a serious public health and economic development challenge. Although there have been a number of studies illustrating considerable sub-national geospatial variation in MN density in key foods, I know of no study that does this at this scale or with this level of precision. This paper should help spark follow on research and can directly inform policy design in several domains. The authors are to be congratulated and thanked for an important contribution.

In the spirit of trying to help further improve a nice piece of work, let me offer a few suggestions for revision. (Note: I cite papers that elaborate on the point merely in the interests of brevity, not to suggest these merit citation.)

Main points:

1. The authors might reflect a bit more on their sampling frame and what it implies for inference. If I understand correctly, they built a sampling frame of locations (i) with a high probability of crop cultivation, based on remote sensing data analysis, and (ii) within 2.5 km of a road. They do not (but should) report what share of the nation's land mass that satisfies (i) gets masked by criterion

(ii) and what share meets both criteria. The sampling frame is thus, loosely speaking, road-accessible cultivated lands. But crop yields (tons/hectare) vary markedly over space, so without weighting for yields, the reported results are averages of the land, not of the cereals supply. The latter would be more meaningful than the former if the authors collected data that permit estimation of local yields. At a minimum, explain more precisely how readers should interpret your claim of 'nationally representative grain surveys' as they do not seem to be representative of grain harvested or consumed if sampled equiproportionately across locations that vary by yield.

2. The authors appropriately emphasize spatial variation in mineral density of key crops. But one naturally wants to know about spatial covariance as well. Are crops likely to be deficient in all minerals if deficient in one (i.e., positive covariances) in a given location, or do relative surpluses of one mineral (partially) compensate for deficiencies in another mineral? The spatial covariance matrix should be readily computable in the data and merits discussion. It matters for how one thinks about interventions to address shortfalls, for example, dietary adjustment over space to rebalance MN availability versus need for supplementation to address multiple deficiencies at once.

3. How much of the variation in sample is within locations (i.e., intra- or inter-farm within the same community) rather than across locations (i.e., geospatial)? The geospatial variation is important. But how much of the variation in crop mineral density does this explain? It might be that other factors are even more important to explaining subnational variation in crop MN density.

4. The analysis stops at cereal grain MN density. The authors need to make clearer to readers that the relationship between crop grain MN density, human MN intake and absorption, and ultimately human MN deficiencies is noisy and conditional on underlying health status (e.g., diarrhea, infectious disease status), other foods in the diet, food preparation methods, use of supplements, etc. Crop MN density is important, especially in places (like rural Ethiopia and Malawi) where lightly processed staple cereals account for a large share of the diet. But boosting crop MN content does not automatically remedy human MN deficiencies. Nor is it necessarily the cheapest approach to do so. The authors clearly know this and hint at it in the concluding section. But a clear caution at the outset could help readers from mistakenly concluding that boosting crop mineral densities is always and everywhere desirable.

5. Following on the prior point, the analysis of dietary contributions relies on a host of herculean assumptions, since the authors lack data on subnational variation in diets, activity levels, disease incidence, etc. that could easily covary with crop mineral densities. This is the weakest part of the paper, extending well beyond the available data. These are back of the envelope calculations, based on coarse nationwide food balance sheets and a single representative estimate average requirement threshold for a single population cohort (women 18-24 years old are important, but a small minority of the local populations). These are not rigorous estimates of dietary requirements. The section on "Mapping cereal grain micronutrient concentrations" should drop or clearly qualify its claims about 'dietary contributions'.

6. This is first and foremost a careful descriptive analysis. But readers naturally want to understand the reasons for the observed geospatial variation, as well as non-spatial (intra-location) variation in MN densities. Other studies (e.g., Bevis and Hestrin forthcoming) have gone a bit further (albeit in smaller, more spatially restricted samples) to establish (surprisingly weak) associations with, for example, soil conditions (pH, soil organic matter concentration) in the plots from which sampled crops were harvested. Are differing MN densities due mainly to crop variety selection, soil conditions, agronomic practices (incl. organic and inorganic fertilizer use), weather/climate, or other factors? Knowing that there is subnational variation is the first step. But what to do about it? For example, if MN density is negatively associated with growing conditions (i.e., yields and MN densities covary negatively) then interventions should be tailored to harvest conditions. Or if MN densities are strongly associated with particular seed varieties or agronomic practices, then extension services (or national input subsidy programs) might account for the follow-on nutritional and health impacts of cultivation practices. If the problem is that poorer

farmers cannot afford to invest in soils, seed and better agronomy, such that there exists reinforcing feedback between farmers' poverty and soil and crop mineral densities (Barrett and Bevis 2015a), then maybe cash transfer programs would be the best remedy?

7. Following on the preceding point, the concluding discussion of implications for food system interventions strikes me as incomplete, perhaps even somewhat off target. More than half of the concluding section focuses on biofortification. While I appreciate the important point that subnational variation in MN density often exceeds breeders' targeted boost in mineral concentrations, it's unclear how this information would effect breeding strategies or extension recommendations. Moreover, multiple entry points exist for remedying MN deficiencies within food systems, not just biofortified germplasm, but equally mineral-enriched fertilizers, industrial fortification, greater dietary diversity, or therapeutic supplements (Barrett and Bevis 2015b). Crop MN density estimates have direct implications for local tailoring of nutrition education, guidance to school feeding managers, and processors/manufacturers that fortify flour or other products to reach a target MN density, among others. The heavy emphasis on biofortification in this last section seems unwarranted.

Minor points:

- i. Figure 3: 'grey shaded area is a mask based on a kriging variance threshold' means nothing to this reader, thus probably to some other readers, too. Perhaps explain more clearly?
- ii. Field sampling: How was sampling adjusted if the sampled location did not contain a target cereal crop (e.g., had a legume or vegetable instead)?
- iii. Field sampling: Samples were taken from the middle of fields. But there exist well-known edge effects in crop yields and these may matter especially for smaller plots cultivated by poorer farmers (Bevis and Barrett 2019).

References

Barrett, C.B. and Bevis, L.E., 2015a. The self-reinforcing feedback between low soil fertility and chronic poverty. *Nature Geoscience*, 8(12), pp.907-912.

Barrett, C.B. and Bevis, L.E., 2015b. The micronutrient deficiencies challenge in African Food Systems. In D.E. Sahn, ed., *The fight against hunger and malnutrition: the role of food, agriculture, and targeted policies*, pp.61-88. Oxford University Press.

Bevis, Leah, Christopher B Barrett. 2019. "Close to the Edge: High Productivity at Plot Peripheries and the Inverse Size-Productivity Relationship," *Journal of Development Economics*, 143(102377).

Bevis, L.E. and R. Hestrin. 2020. Variation in crop zinc concentration influences estimates of dietary Zn inadequacy. *PloS one*, 15(7), p.e0234770.

Bevis, L.E. and R. Hestrin, Widespread heterogeneity in staple crop mineral concentration in Uganda partially driven by soil characteristics, *Environmental Geochemistry and Health*, Forthcoming.

Author Rebuttals to Initial Comments:

Referee 1	Response
1	As an agronomist, I would not call Ca a micronutrient. In my view it is a macronutrient, at least for a plant. We appreciate that Ca is a macronutrient in plant nutrition field, but it is classed as a micronutrient in the study of human nutrition. We have clarified terminology by stating "among people" in the first sentence of the Abstract.

2	Variation in soil acidification might play a major role in this data set. At least for Ethiopia I have seen soil maps with huge variation in pH levels on short distances which will obviously have a huge impact on uptake of certain micronutrients by the crop. Is it possible to overlay the variation in quality with the soil pH maps?	See response to Editor. We have included a new geostatistical analysis of the influence of soil pH on grain Se and Zn concentration.
3	Similarly, there is huge variation in altitude in Ethiopia, also over relatively short distances. It is likely that variation in elevation will have strong effects on micronutrient content because of variation in temperature during grain filling. Temperature will strongly increase grain filling rate and shorten grain filling duration, with the ultimate effect that single-grain weight will become lower when temperature during grain filling will be higher. It would be useful to assess whether the variation in micronutrient content is associated with variation in altitude.	See response to Editor. We have included a new geostatistical analysis of the influence of environmental covariates on grain Se and Zn concentration.
4	Related with point 3 but also related to general agronomic conditions: Thousand grain weight could be an important explaining co-variable both among species and within species. Has that trait been recorded? From a crop physiological and agronomic point of view it would be essential to interpret variation in micronutrient content based on that trait, so the background of the geospatial variation would be much easier interpreted when that trait is known and reported.	The influence of agronomy and crop genotype were not determined in this study. These factors are important determinants of yield and yield components, which will in turn influence grain micronutrient concentration. We have added new text stating the need to consider G × E × M sources of variation in the final paragraph of the main text (P9).
5	How does the processing of the grain samples in this research compare with the way the grains are processed for food? Could that be discussed in more detail?	We used whole-grains, which is similar to preparation used in some food preparation steps. We have added clarification text in the Materials and Methods, "Sample Preparation" section (P16).

6	Are there enough data collected for a high level of accuracy in Figures 3 and 4? Contrasts in these figures are often very large over very short distances and I wondered whether that had to do with small numbers of observations for specific crops in specific parts of the country.	Any spatial predictions have attendant uncertainty, and this should be considered when interpreting them. The outputs of Figures 3 and 4 should be interpreted in combination with the kriging variance figures, which indicate the expected squared prediction error. We have added new text to flag this (P5, "Mapping..." paragraph).
7	Diet diversification is probably a faster route to solving hidden hunger than breeding for increased micronutrient levels. This could be discussed in more detail.	We now discuss diet diversification, including potential effects on livestock health and production, alongside potential interventions, with new text added (penultimate paragraph, P9).
8	The statement in lines 263 – 265 on the decline in micronutrient concentration caused by the impact of increased CO₂ in the future is questionable. There have been some alarming papers on that potential effect but in many cases the other factor in climate change (increased temperature) was not taken into account. The increase in temperature compensates for the increase in CO₂ and the outcome of those opposite effects is uncertain and very much depends on the crop and on the micronutrient. We have collected detailed data sets from T-FACE experiments and we could not see a consistent effect on micronutrient contents in the grain of wheat and rice nor in their bioavailability (paper submitted, not published yet). Other research have also reported on micronutrient contents in wheat, rice and soybean grain obtained from T-FACE experiments and concluded that elevated temperature could compensate for elevated CO₂. See e.g. Köhler, I. H., Huber, S. C., Bernacchi, C. J., & Baxter, I. R. (2019). Increased temperatures may safeguard the nutritional quality of crops under future elevated CO₂ concentrations. The Plant Journal, 97(5), 872-886. Wang, J., Li, L., Lam, S. K., Liu, X., &	We agree with this. We now report the positive association between temperature and grain Se and Zn concentration as part of the new geostatistical analyses. We have added discussion text and cited the Köhler et al. paper in the final paragraph of the main text (P9).

	Pan, G. (2020). Responses of wheat and rice grain mineral quality to elevated carbon dioxide and canopy warming. Field Crops Research , 249, 107753.	
9	It is highly likely that better grain yields are associated with lower micronutrient concentrations, but nevertheless, higher yields would allow more food intake and therefore more micronutrient intake despite lower concentrations. Can the authors discuss this as well and can they speculate to what extent the geospatial data on micronutrients are in line with or in contrast to geospatial variation in yield?	See Reviewer 1, Response 4.
	Typo line 404: farmers should be farmers Lines 804-805: sentence incomplete.	Corrected.
Referee 2		Response
1	Explanatory value: The study presents data in a descriptive way and identified spatial dependencies acting over a maximal distance but does not provide more insights in what drives micro-nutrient concentrations. While the authors took paired grain-soil samples, there is no further mention of the soil information in the paper. We also know that in both countries, fertilizer use is happening with some of these products having Ca and/or Zn included. The impact of this management intervention is not presented. In short, my feeling is that the presented work is a really excellent and necessary first step but more work is needed to add value to our understanding of what drives micro-nutrient concentrations.	See response to Editor. We have included a new geostatistical analysis of the influence of soil and environmental covariates on grain Se and Zn concentration. In the areas sampled, there is very limited use of micronutrient fertilisers that would influence the grain micronutrient concentration. However, initiatives are underway in both countries to promote micronutrient fertilisers. This work provides the basis for more targeted hypothesis testing of the effect of yield and yield components on grain micronutrient concentration. We have added new text stating the need to consider $G \times E \times M$ sources of variation in the final paragraph of the main text (P9). As flagged by the other referees, this is an essential first step to addressing the missing evidence at this spatial scale.
2	Varietal differentiation: We know that different varieties of the targeted cereals very likely have varying micro-nutrient uptake	We agree fully and we have added new text, stating the need to consider $G \times E \times M$ sources of variation, in the final paragraph of the main text (P9).

	potentials and this is not explored by the authors. It would be key to zoom in on this issue, which could explain some of the unexplained observed variation.	
3	Variation over time: It would be good to understand how seasonal variations or variations in crop and soil management practices over time affect micro-nutrient concentrations. It is likely that concentrations are not constant over time.	This is also an important issue, however, the current study is limited to the main harvest period in both countries.
4	Linkages between measured micro-nutrient concentrations and dietary supply (Figs 3 and 4): Estimation of the dietary supply is done in a 'quick and dirty' way and does not include information on total availability of grain, intake of other food groups, nor the bio-availability of the micro-nutrients for the different crops. Therefore, in their current form, the information presented is of limited value and certainly not sufficient for policy formulation.	See response to Editor. We have addressed this comment by conducting new spatial modelling to provide evidence that the variation in grain micronutrient composition is linked to a dietary outcome. This type of analysis is currently only possible for Se because direct evidence of dietary micronutrient quality for Ca, Fe, and Zn—from biomarkers of micronutrient status and dietary survey data (i.e. recall data, with single food composition data points for each food type)—are not yet adequate for such a purpose.
Referee 3		Response
1	This manuscript reports an enormous body of work which is quite staggering in its coverage – the coordination and logistical effort that has been taken to collect and process such a large number of samples from such a wide geographic areas in both Ethiopia and Malawi is truly impressive. The sampling frame and statistical approach used to locate the samples and the methods used for sampling and analysis look to be well thought through and robust. The data analysis and presentation is clear, although the figures and maps are too small to read the detail. I suggest that large versions of all maps are published online to allow detailed scrutiny of the maps by readers.	We thank the reviewer for these comments. We have included thumb-nail figures in the submitted text for ease of reviewing. Full-scale, high-resolution maps have been submitted alongside the paper and these will be published online.

	My concern is whether the authors have truly established that, as stated in lines 46-47, the “geospatial variation ... is nutritionally important”. This point is discussed in the paragraph from lines 197-209. We are told that the areas where larger concentrations of selenium are found in the grain co-locate with areas where better Se status of children has been reported. So essentially the paper relies on previously published information to state that the geographical patterns of micronutrient distribution are meaningful for human nutrition and health. As a reader there appears to be no way to check this – the paper certainly has demonstrated large differences in micronutrient content of cereal grains across geographic areas but I wonder whether the authors have any means to demonstrate more clearly the links with human nutrition.	See response to Editor. We have addressed this comment by conducting new spatial modelling to provide evidence that the variation in grain micronutrient composition is linked to a dietary outcome.
3	It is not clear what the authors are recommending in terms of “surveillance” – would it be sufficient to conduct a one time national survey such as this to establish which regions are of concern in terms of restricted concentrations of micronutrients in grain? Or would more regular sampling be needed? Given the huge amount of work involved to conduct such a spatial analysis it seems unlikely this would be feasible on any regular basis. Further all samples were shipped to the UK for analysis – is such analysis feasible within each country or would substantial investments be needed in laboratory infrastructure and training?	We have added clarification text on laboratory access (first paragraph of “Implications...” section on P9). We have also explained more clearly how a crop micronutrient survey within a country could inform interventions to alleviate MNDs (penultimate paragraph, P9).
4	Little is made of the spatial dependence of concentrations of different nutrients in grain. If these are related to soil type – why is there different spatial dependence for	Our new analyses of the grain nutrient concentrations [Response to Editor and earlier referees’ comments] show that different micronutrients in different crops do not, in general, have the same relationship to soil

	different nutrients? Surely this is one of the more interesting outcomes of the study? Are the differences among soil types masked by differences in management (use of organic and mineral fertilisers?). It is well established that soil fertility varies enormously over very short distances (50-100 m) within farms in sub-Saharan Africa and in particular in Ethiopia there are strong catenary effects on soil type/ soil fertility over distances of a few km. These are very different scales compared with those mapped at subnational scale and discussed in the paper (see L 183-195).	properties. For example, in Ethiopia, wheat Zn concentration was significantly related to soil organic carbon (SOC) content (positive relationship) but not to pH; teff Zn concentration was significantly related to pH (positive relationship) but not to SOC, while both of these soil properties were significantly related to the Zn concentration in maize grain (positive relationships). Given this, as well as different relationships to environmental covariates, the spatial dependence of the grain nutrient concentrations differs between crops. We have added new text, stating the need to consider $G \times E \times M$ sources of variation, in the final paragraph of the main text (P9). The key finding for this paper is that the supply in the staple crop is not only variable, but correlates across distances of 10s–100s of km. Further work, e.g. modelling soil-to-crop-diet transfers could in future show how interventions to alleviate MNDs might be targeted to local conditions.
	I find the final paragraph of the paper is rather weak and needs rethinking to emphasise the importance of this study.	We have substantially revised the final two paragraphs (P9).
MINOR	Please make it clear from the start that the article is about human nutrition and not plant nutrition L 47 – representative – of what? Spatially representative? – this is defined later in L 117 as “systematic spatial sampling” but this needs to be defined earlier. L 55 – in what way are food fortification and biofortification “food system interventions”? “geographical effects which can be larger than planned intervention outcomes” L 61 – greater than what? L 80 – in the human body?	Clarified in first sentence of Abstract. We have removed the word “representative” from the Abstract, to clarify it is national scale. We have clarified the sampling domains elsewhere in the paper. We have clarified text. Clarified. Clarified.

	L 88 delete “having” L 91 – single values from tables – these are not usually single data points? L 91 – confusing punctuation L 96 do not use “a.k.a.” – delete or define “geonutrition’ – this is not a commonly used term and unnecessary jargon? L 96/97 – is this true? The paper cites several studies that show such links – the statement needs clarification L 109 – nationally representative? See comment above. L 120 – composite not composited L 205 – vertisols are NOT alluvial soils – this needs correcting!! Vertisols are common in Ethiopia but much less common in Malawi L 264 – there is considerable debate as to whether elevated CO₂ will reduce cereal grain micronutrient concentrations due to the compensating effects of increased temperatures. This statement needs more nuance. L 423 – remote sensing? L 434 – turn around “is generally better to cropped areas” to read “to cropped areas is generally better” L 530 – in this paper? Rather than study. Figs 3 and 4 – the keys are too small to read. I do not see a grey mask in Fig. 4?	Deleted. Clarified. Clarified. Deleted. Clarified. Clarified [See Referee 4, Comment 1]. Changed. Deleted ‘alluvial’. Edited [See Referee 1, Response 8]. Changed. Edited. Clarified to state that soil pH and SOC are reported in this study/ Full-size figures will be published online and have been provide here for review. The reference to grey-shaded area has been clarified in the legend to Fig. 3 and removed
--	---	--

		from the legend to Fig. 4. [See Referee 3, Response 1].
Referee 4		Response
1	The authors might reflect a bit more on their sampling frame and what it implies for inference. If I understand correctly, they built a sampling frame of locations (i) with a high probability of crop cultivation, based on remote sensing data analysis, and (ii) within 2.5 km of a road. They do not (but should) report what share of the nation's land mass that satisfies (i) gets masked by criterion (ii) and what share meets both criteria. The sampling frame is thus, loosely speaking, road-accessible cultivated lands. But crop yields (tons/hectare) vary markedly over space, so without weighting for yields, the reported results are averages of the land, not of the cereals supply. The latter would be more meaningful than the former if the authors collected data that permit estimation of local yields. At a minimum, explain more precisely how readers should interpret your claim of 'nationally representative grain surveys' as they do not seem to be representative of grain harvested or consumed if sampled equiproportionately across locations that vary by yield.	We have substantially clarified the text around the statement “nationally representative grain surveys” throughout. This includes edited text in the “Grain Surveys...” section (P3), and “Sampling Design” section (P14), which includes an estimate of land area meeting the criterion for selection in Ethiopia. This survey design did not permit us to obtain yield measurements, which we have included as a note as being important for future targeted studies. We have added new text, stating the need to consider $G \times E \times M$ sources of variation affecting yield and yield components, in the final paragraph of the main text (P9).
2	The authors appropriately emphasize spatial variation in mineral density of key crops. But one naturally wants to know about spatial covariance as well. Are crops likely to be deficient in all minerals if deficient in one (i.e., positive covariances) in a given location, or do relative surpluses of one mineral (partially) compensate for deficiencies in another mineral? The spatial covariance matrix should be readily computable in the data and merits discussion. It matters for	See Referee 3, Response 4. Different micronutrients in different crops do not, in general, have the same relationship to soil properties. Further multivariate geostatistical analysis of these data would have to be done in a model-based way (it is not simply a matter of computing a covariance matrix), due to the non-random sampling design. Such analyses are beyond the scope of this paper.

	how one thinks about interventions to address shortfalls, for example, dietary adjustment over space to rebalance MN availability versus need for supplementation to address multiple deficiencies at once.	
3	How much of the variation in sample is within locations (i.e., intra- or inter-farm within the same community) rather than across locations (i.e., geospatial)? The geospatial variation is important. But how much of the variation in crop mineral density does this explain? It might be that other factors are even more important to explaining subnational variation in crop MN density.	These uncertainties (i.e. $G \times E \times M$ components which were not measured in this study) are now described in much more detail. We have added new text, stating the need to consider $G \times E \times M$ sources of variation, in the final paragraph of the main text (P9).
4	The analysis stops at cereal grain MN density. The authors need to make clearer to readers that the relationship between crop grain MN density, human MN intake and absorption, and ultimately human MN deficiencies is noisy and conditional on underlying health status (e.g., diarrhea, infectious disease status), other foods in the diet, food preparation methods, use of supplements, etc. Crop MN density is important, especially in places (like rural Ethiopia and Malawi) where lightly processed staple cereals account for a large share of the diet. But boosting crop MN content does not automatically remedy human MN deficiencies. Nor is it necessarily the cheapest approach to do so. The authors clearly know this and hint at it in the concluding section. But a clear caution at the outset could help readers from mistakenly concluding that boosting crop mineral densities is always and everywhere desirable.	See response to Editor. We have addressed this comment by conducting new spatial modelling to provide evidence that the variation in grain micronutrient composition is linked to a dietary outcome. We have included additional text on other factors driving micronutrient deficiency, including the need for a diverse diet, in the penultimate paragraph of the main text (P9).
5	Following on the prior point, the analysis of dietary contributions relies on a host of herculean	See response to Editor. We have addressed this comment by conducting new spatial modelling to provide evidence that the variation in grain

	assumptions, since the authors lack data on subnational variation in diets, activity levels, disease incidence, etc. that could easily covary with crop mineral densities. This is the weakest part of the paper, extending well beyond the available data. These are back of the envelope calculations, based on coarse nationwide food balance sheets and a single representative estimate average requirement threshold for a single population cohort (women 18-24 years old are important, but a small minority of the local populations). These are not rigorous estimates of dietary requirements. The section on "Mapping cereal grain micronutrient concentrations" should drop or clearly qualify its claims about 'dietary contributions'.	micronutrient composition is linked to a dietary outcome. We have clarified that the use of a threshold, based on an adult woman aged 18–24 years eating an unrefined (i.e. high phytate) diet is representative (P5). In the Methods (P19), we have noted that the thresholds will be similar for other demographic groups.
6	This is first and foremost a careful descriptive analysis. But readers naturally want to understand the reasons for the observed geospatial variation, as well as non-spatial (intra-location) variation in MN densities. Other studies (e.g., Bevis and Hestrin forthcoming) have gone a bit further (albeit in smaller, more spatially restricted samples) to establish (surprisingly weak) associations with, for example, soil conditions (pH, soil organic matter concentration) in the plots from which sampled crops were harvested. Are differing MN densities due mainly to crop variety selection, soil conditions, agronomic practices (incl. organic and inorganic fertilizer use), weather/climate, or other factors? Knowing that there is subnational variation is the first step. But what to do about it? For example, if MN density is negatively associated with growing conditions (i.e., yields and MN densities covary negatively) then interventions should be tailored to harvest conditions. Or if MN densities are strongly	See response to Editor. We have now included detailed geostatistical analysis of the influence of soil properties and potential environmental co-variables on variation in grain micronutrient concentration. We agree there is a need for much more detailed exploration of $G \times E \times M$ components, which will be highly context specific for different geographical locations, farming systems, climates, etc. Similarly, the policy responses will also need to be context-specific. We hope that our new discussion text captures this complexity much better in the final two paragraphs (P9). We thank the reviewer for the reference suggestions and have included the following as important studies which needed citing: [Ref. 17] Bevis, L. E. & Hestrin, R. Widespread heterogeneity in staple crop mineral concentration in Uganda partially driven by soil characteristics. Environmental Geochemistry and Health in press (2020). https://doi.org/10.1007/s10653-020-00698-w. [Ref. 36] Barrett, C. B. & Bevis, L. E. The self-reinforcing feedback between low soil fertility

	associated with particular seed varieties or agronomic practices, then extension services (or national input subsidy programs) might account for the follow-on nutritional and health impacts of cultivation practices. If the problem is that poorer farmers cannot afford to invest in soils, seed and better agronomy, such that there exists reinforcing feedback between farmers' poverty and soil and crop mineral densities (Barrett and Bevis 2015a), then maybe cash transfer programs would be the best remedy?	and chronic poverty. Nature Geoscience 8, 907–912 (2015). https://doi.org/10.1038/ngeo2591.
7	Following on the preceding point, the concluding discussion of implications for food system interventions strikes me as incomplete, perhaps even somewhat off target. More than half of the concluding section focuses on biofortification. While I appreciate the important point that subnational variation in MN density often exceeds breeders' targeted boost in mineral concentrations, it's unclear how this information would effect breeding strategies or extension recommendations. Moreover, multiple entry points exist for remedying MN deficiencies within food systems, not just biofortified germplasm, but equally mineral-enriched fertilizers, industrial fortification, greater dietary diversity, or therapeutic supplements (Barrett and Bevis 2015b). Crop MN density estimates have direct implications for local tailoring of nutrition education, guidance to school feeding managers, and processors/manufacturers that fortify flour or other products to reach a target MN density, among others. The heavy emphasis on biofortification in this last section seems unwarranted.	We have re-written the final two paragraphs to capture the complexity more appropriately and to avoid the over-focus on biofortification (P9).

8	i. Figure 3: 'grey shaded area is a mask based on a kriging variance threshold' means nothing to this reader, thus probably to some other readers, too. Perhaps explain more clearly? ii. Field sampling: How was sampling adjusted if the sampled location did not contain a target cereal crop (e.g., had a legume or vegetable instead)? iii. Field sampling: Samples were taken from the middle of fields. But there exist well-known edge effects in crop yields and these may matter especially for smaller plots cultivated by poorer farmers (Bevis and Barrett 2019).	(i) Text clarified in Methods (P19, second paragraph), with a note included in the Figure 3 legend. (ii) Field teams went to nearest cereal field. Text clarified in Methods ("Field Sampling", P15). (iii) a discussion of the variation operating over smaller distances now included in the final paragraph (P9). We thank the reviewer for reference suggestion and have included: [Ref. 37] Bevis, L. E. M. & Barrett, C. B. Close to the edge: High productivity at plot peripheries and the inverse size-productivity relationship. Journal of Development Economics 143, 102377 (2020). https://doi.org/10.1016/j.jdeveco.2019.102377.
---	---	--

Reviewer Reports on the First Revision:

Referee #1 (Remarks to the Author):

The authors have addressed all my comments in a satisfactory way. I am happy that they took up the challenge as some of my comments required significant additional analyses.

I also checked how the authors dealt with the comments of the editor and the other reviewers and in my opinion they have overall done a very good job in addressing all the comments and writing a very clear rebuttal letter.

In my view this significant paper can be accepted for publication.

Referee #2 (Remarks to the Author):

Comments from the reviewers have been addressed with great care and I recommend the paper for acceptance.

Referee #3 (Remarks to the Author):

I have read the revised manuscript and the authors response to the referees' comments. The authors have taken all comments seriously and addressed them well. They have conducted extra spatial modelling to address the issue of evidence of the dietary contribution that the spatial variation in micronutrient concentration in grains can make.

I think the authors have gone as far as they can to demonstrate this - and I do not think any further analysis is needed. Nevertheless I think that authors need to be a little more circumspect in the conclusions that they make.

Firstly, the link between grain content and a human biomarker can be made for selenium but not for the other nutrients. The authors should be clear about this - that a direct link is demonstrated for Se, and that the variations in concentration for the other micronutrients are large enough to assume that they will be important in terms of dietary intake. I think the authors need to be clear on this in the paper - and that further research would be needed to confirm the importance for the human diet.

Second, the soil variables that are important in determining variation in grain nutrient concentration have been investigated for the various micronutrients. The relationships are consistent for Se for all crops in both countries. However for Zn they are not consistent - being positive in some case and negative in others. The authors do not discuss this either in terms of the causes for the inconsistent relationships not in terms of what that means for how their results can be used. The lack of consistent relationships could be (partly) due to the spread of values for pH in Ethiopia which is much wider than in Malawi where acid soils ($\text{pH} < 5$) are rarer. But it does indicate that the predictive value of soil values is weak. I find the paragraph (lines 225-238) of the manuscript unsatisfactory as it simply states these relationships with no explanation or discussion of their importance.

Referee #4 (Remarks to the Author):

Thanks for the significant new work done to connect the soil micronutrient estimates to nutritional status. That is an important extension. Likewise, the clarifications on sampling nicely addressed my confusion from the prior version.

My only quibble is that your response did not really address my prior comment 2. One could readily compute the spatial covariances in crop micronutrient densities; if you can compute the spatial variance for a given micronutrient you can quite easily compute the covariances for different micronutrients drawn from the same crop samples. If the non-random sampling design confounds the covariances then it also confounds the variances that are at the heart of the present analysis. That does not need a structural model. You are of course correct that different micronutrients respond differently to different soil properties. But soil properties covary significantly across space, so it is reasonable to hypothesize that crop micronutrient densities might as well, albeit due to different biochemical mechanisms. But this matters for the targeting of interventions to address human micronutrient deficiencies. The manuscript is already rich with important new findings. But it seems a missed opportunity not to explore these covariances, especially when advocating for surveillance systems for whose optimal design - e.g., sample sizes needed for adequate statistical power - depend on those covariances. I defer to the editor and authors whether that is best left for a different manuscript or as a brief addition to this one.

Nice work. This was a pleasure to read.

Author Rebuttals to First Revision:

There are two minor comments to address from Referee 3; our response in red:

1. **“the variations in concentration for the other micronutrients are large enough to assume that they will be important in terms of dietary intake ... [but] ... further research would be needed to confirm the importance for the human diet”.** We have added a sentence in the results (L281), “Direct evidence of linkages between the grain concentration of other micronutrients, biomarkers of dietary status, and health outcomes remains a major research challenge.”
2. **Relationships between soil/environmental covariates and grain Zn concentration are not consistent.** Relationships between soil organic carbon and grain Zn concentration, and between environmental covariates and grain Zn concentration are consistent across the different crops/countries. Relationships between soil pH and grain Zn concentration do differ between teff (negative) and maize (positive) in Ethiopia. Maize grain Zn concentration is also positively related to soil pH in Malawi. This indicates a crop-specific phenomenon. We have added a clarifying sentence (L290), “Further studies of contrasting responses to soil pH, observed in teff and maize, are needed. However, the generally weak predictive value of soil pH on grain Zn concentration (Extended Data Figures 7, 9) is consistent with a survey of staple crops in Uganda¹⁷.”

There is one minor comment to address from Referee 4; our response in red:

1. **Editorial steer on the issue of spatial covariances in crop micronutrient densities.** We understand the reviewer's opinion that estimating covariances should be as straightforward as estimating variances. For method-of-moments estimation from data obtained by independent random sampling that is certainly true. Non-independent sampling does not "confound" covariances or variances, but it does mean that we must then use model-based methods to estimate them, and these introduce more restrictive assumptions in the multivariate case than in the univariate case. This is discussed, for example, by Webster and Oliver in Chapter 10 of their "Geostatistics for Environmental Scientists" (Methods Reference 61). We certainly agree that a multivariate spatial analysis would be informative, as would be the more general issue of optimising the design of wider surveillance systems. However, we believe this would be well-beyond the scope of this current study. We have added the following text (L313), “Multivariate spatial statistical modelling could be informative about soil and environmental factors which jointly influence the variation in grain micronutrient concentration. However, such modelling requires extensive assumptions which are outside the scope of this study.”